# An ATR-PrimPol pathway confers tolerance to oncogenic KRAS-induced and heterochromatin-associated replication stress

Taichi Igarashi[1,2], Marianne Mazevet[1], Takaaki Yasuhara [3], Kimiyoshi Yano[1], Akifumi Mochizuki[4,5], Makoto Nishino[4], Tatsuya Yoshida[6], Yukihiro Yoshida[7], Nobuhiko Takamatsu[2], Akihide Yoshimi [2,8], Kouya Shiraishi [4,9], Hidehito Horinouchi [6], Takashi Kohno [4], Ryuji Hamamoto [10], Jun Adachi [11], Lee Zou[12,13,14] & Bunsyo Shiotani [1] ✉

Activation of the *KRAS* oncogene is a source of replication stress, but how this stress is generated and how it is tolerated by cancer cells remain poorly understood. Here we show that induction of KRAS^G12V expression in untransformed cells triggers H3K27me3 and HP1-associated chromatin compaction in an RNA transcription dependent manner, resulting in replication fork slowing and cell death. Furthermore, elevated ATR expression is necessary and sufficient for tolerance of KRAS^G12V-induced replication stress to expand replication stress-tolerant cells (RSTCs). PrimPol is phosphorylated at Ser255, a potential Chk1 substrate site, under KRAS^G12V-induced replication stress and promotes repriming to maintain fork progression and cell survival in an ATR/Chk1-dependent manner. However, ssDNA gaps are generated at heterochromatin by PrimPol-dependent repriming, leading to genomic instability. These results reveal a role of ATR-PrimPol in enabling precancerous cells to survive KRAS-induced replication stress and expand clonally with accumulation of genomic instability.

Genomic instability is a hallmark of cancer and is a driving force for the acquisition of other characteristics of cancer[1]. In hereditary cancers, mutations in DNA repair genes cause genomic instability that promotes cancer development. On the other hand, recent high-throughput sequencing studies have revealed that sporadic cancers have fewer defects in DNA repair genes[2]. Instead, more than half of the top 20 genes mutated in human cancers are associated with features of sustained proliferation, raising the possibility that genes driving cell proliferation, such as oncogenes, are embracing complex environmental changes that disrupt cellular processes and contribute to genomic instability[3]. Such cellular disruptions involve physiological dysregulation of DNA replication and are widely referred to as

replication stress (RS). A large body of evidence indicates that during the early stages of tumorigenesis, oncogenes cause RS, leading to genomic instability[4]. While replication cannot be completed in most of these distressed cells, resulting in programmed cell death or senescence[5], a limited number continue to proliferate and expand clonally in noncancer tissues[6]. These findings raise the question of how cells adapt to RS while driving genomic instability during early carcinogenic processes.

*KRAS* is one of the most frequently mutated oncogenes in human cancers and is considered to be an important early driver of many tumors[7]. *KRAS* mutations commonly lead to substitution of a single amino acid (G12) in the KRAS protein with another amino acid, which

causes constitutive activation of the downstream RAS signaling cascade and induces oncogenic events[8,9]. For instance, such mutations cause loss of normal growth barriers and disruption of tissue homeostasis resulting in increased exposure of mutant KRAS cells to stress-inducing conditions such as RS[10,11]. Recent evidence suggests that strong overexpression of oncogenic Ras in normal fibroblasts increases RNA synthesis accompanied by R-loop accumulation, which interferes with replication fork progression and causes RS, whereas inhibition of origin firing failed to abrogate RAS-induced RS in the same system[12]. However, the obstacles to abrogating fork progression have not been fully addressed, and whether a similar mechanism is active in normal human epithelial cells expressing oncogenic KRAS mutants—a frequent situation in human carcinogenesis—remains elusive.

Ataxia Telangiectasia and Rad3-related kinase (ATR), a master regulator of RS, and its critical substrate, Checkpoint Kinase 1 (Chk1), regulate cell cycle checkpoints and facilitate DNA repair via their substrates[13–19]. Therefore, ATR and Chk1 have long been considered tumor suppressors[20–22]. However, recent studies have shown that ATR and Chk1 are required for cell survival in response to oncogenic RAS expression during tumorigenesis, suggesting that the ATR-Chk1 kinase pathway protects cells from deleterious and chronic RS induced by oncogenes, thus promoting RS tolerance (RST)[23–26]. As a mechanism of RST in human cells, repriming of DNA synthesis at stalled replication forks promotes discontinuous replication at leading strands with single-stranded DNA (ssDNA) gaps[27,28]. Human primase and DNA-directed polymerase (PrimPol) is a major factor that mediates the repriming process during the RS response[29]. PrimPol needs to be tightly regulated during DNA replication to prevent aberrant repriming, fork speeding, and chromosomal breakage, which increase the risk of genomic instability[30–33]. Given the ability of ATR to protect cells from chronic RS induced by oncogenes and its potential role in regulating repriming, we sought to understand how ATR affects the RS response and determine the role of ATR-mediated RST during cancer development.

Here, we report that KRAS[G12V] regulates chromatin dynamics by triggering H3K27me3 in an RNA transcription-dependent manner and generates HP1-associated chromatin compaction, slowing replication forks and resulting in cell death. In surviving cells with KRAS[G12V] expression, ATR expression increases through suppression of miR185-mediated posttranscriptional regulation. Mechanistically, ATR overexpression leads to unrestrained DNA replication and RST via PrimPol-mediated repriming, which is regulated by ATR/Chk1 kinase activity-dependent phosphorylation of PrimPol at Ser255. We also show that ssDNA gaps are generated near heterochromatin associated with H3K27me3 and HP1 by PrimPol-dependent repriming, leading to genomic instability, which is pronounced in RS-tolerant cells (RSTCs) with high expression of ATR. These findings may underlie the poor prognosis of lung cancer patients harboring KRAS mutations and high ATR expression.

## Results

### The ATR level correlates with aggressive LUAD phenotypes
The ATR-Chk1 pathway is required for cell survival in response to oncogenic RAS expression during tumorigenesis and controls tumor progression in a dosage-dependent manner in mouse models[24], but how this pathway modulates lung adenocarcinoma (LUAD) phenotypes under KRAS-induced RS is largely unknown. To determine whether ATR overexpression could impact KRAS-driven cancer development, we focused on a cohort of LUAD patients from The Cancer Genome Atlas (TCGA) and sought to determine whether the expression level of ATR impacts overall survival (OS). Remarkably, among patients with KRAS-mutant tumors but not among those with KRAS wild-type tumors, patients with high ATR expression (55.4% of patients with KRAS-mutant and 17.4% of all LUAD patients) had a significantly lower OS rate than those with low ATR expression (Fig. 1a). Similar trends were observed in pancreatic adenocarcinoma (PAAD) and colon

adenocarcinoma (COAD) (Supplementary Fig. 1a) although these were not statistically significant. Interestingly, pan-cancer cohort analysis indicated that KRAS-mutant patients with high ATR expression (389/907, 42.9%) had a significantly lower OS rate than patients with low ATR expression (Supplementary Fig. 1b). These data indicate that high ATR expression promotes cellular fitness in KRAS-mutant tumors exposed to RS and is associated with an aggressive phenotype despite the enhanced genotoxic stress.

### Elevated ATR expression promotes cell survival in the presence of KRAS[G12V] expression
To investigate the mechanisms involved in the RS response when oncogenic KRAS is expressed in epithelial cells, we utilized small airway epithelial cells (SAECs) immortalized by transduction of h-TERT, CyclinD1 and a CDK4[R24C] mutant without p16INK4A-binding ability to bypass senescence. These immortalized SAECs express lung differentiation markers and have intact p53[34], confirming that they retain the characteristics of primary SAECs. We retrovirally established SAECs with 4-hydroxytamoxifen (4OHT)-inducible expression of KRAS[G12V] (Fig. 1b and Supplementary Fig. 1c). In most SAECs expressing KRAS[G12V], macropinocytosis was induced, with accumulation of large-phase lucent vacuoles in the cytoplasm, subsequently leading to a form of cell death called methuosis (Fig. 1b–d)[35]. However, a few cells survived and slowly proliferated (Supplementary Fig. 1d), and ATR expression started to increase on day 14 and remained higher than the basal level until day 28 without appreciable activation during this process (Fig. 1e and Supplementary Fig. 1e). After up to 35 days of 2D culture, SAECs expressing KRAS[G12V] showed a limited but significant anchorage-independent growth (Fig. 1f). We picked up these individual clones and named them RS-tolerant cells (RSTCs, see below) (Fig. 1g), which might mimic clonally expanded noncancer cells harboring mutations in cancer driver genes[6]. Interestingly, all clones (RSTC#2, #5, and #7) induced an increase in ATR protein expression compared with that in control cells (Fig. 1h). In RSTCs, ATR mRNA expression was increased more than twofold (Supplementary Fig. 1f), suggesting the involvement of microRNA (miRNA)-mediated posttranscriptional regulation. Recent studies have reported that oncogenic KRAS induces lung tumorigenesis through miRNA modulation[36–38] and that miR-185 suppresses ATR expression via posttranscriptional regulation by binding to the 3′ untranslated region (3′-UTR) of ATR mRNA[39]. Our data showed that miR185 expression was decreased in all RSTCs, supporting the model proposing that increased ATR expression is due at least partially to downregulation of miR185 in cells expressing KRAS[G12V] (Supplementary Fig. 1g). In addition, whole-genome sequencing (WGS) revealed a sporadic copy number gain in the ATR gene in RSTCs (Supplementary Fig. 1h). Furthermore, the ATR protein exhibited higher expression and a longer half-life in RSTCs despite the nearly identical proportion of S-phase cells (Supplementary Fig. 1i, j). In the presence of a CDK inhibitor (roscovitine (Rosc)), the ATR protein expression level in RSTCs decreased but was maintained at a level comparable to that in cycling control cells (Supplementary Fig. 1k). These multilayered mechanisms contributed to maintaining the elevated ATR levels in RSTCs even when KRAS[G12V] induction was turned off, suggesting that the increased ATR level in RSTCs is a stable outcome (Supplementary Fig. 1l). Next, to determine whether forced expression of ATR in normal SAECs enhances survival in the presence of KRAS[G12V] expression, we generated SAEC cells constitutively expressing ATR (hereafter termed ATR-1 and ATR-2 cells) (Fig. 1i) and confirmed that the proliferation of these cells was restored (Supplementary Fig. 1m). Accordingly, ATR-1 and ATR-2 cells expressing KRAS[G12V] formed greater numbers of anchorage-independent colonies than control SAECs (Fig. 1j). Moreover, KRAS[G12V] expression induced an epithelial-mesenchymal transition (EMT) phenotype, including a decrease in E-cadherin and an increase in vimentin expression, during 2D culture (Supplementary Fig. 1n) and in RSTC clones

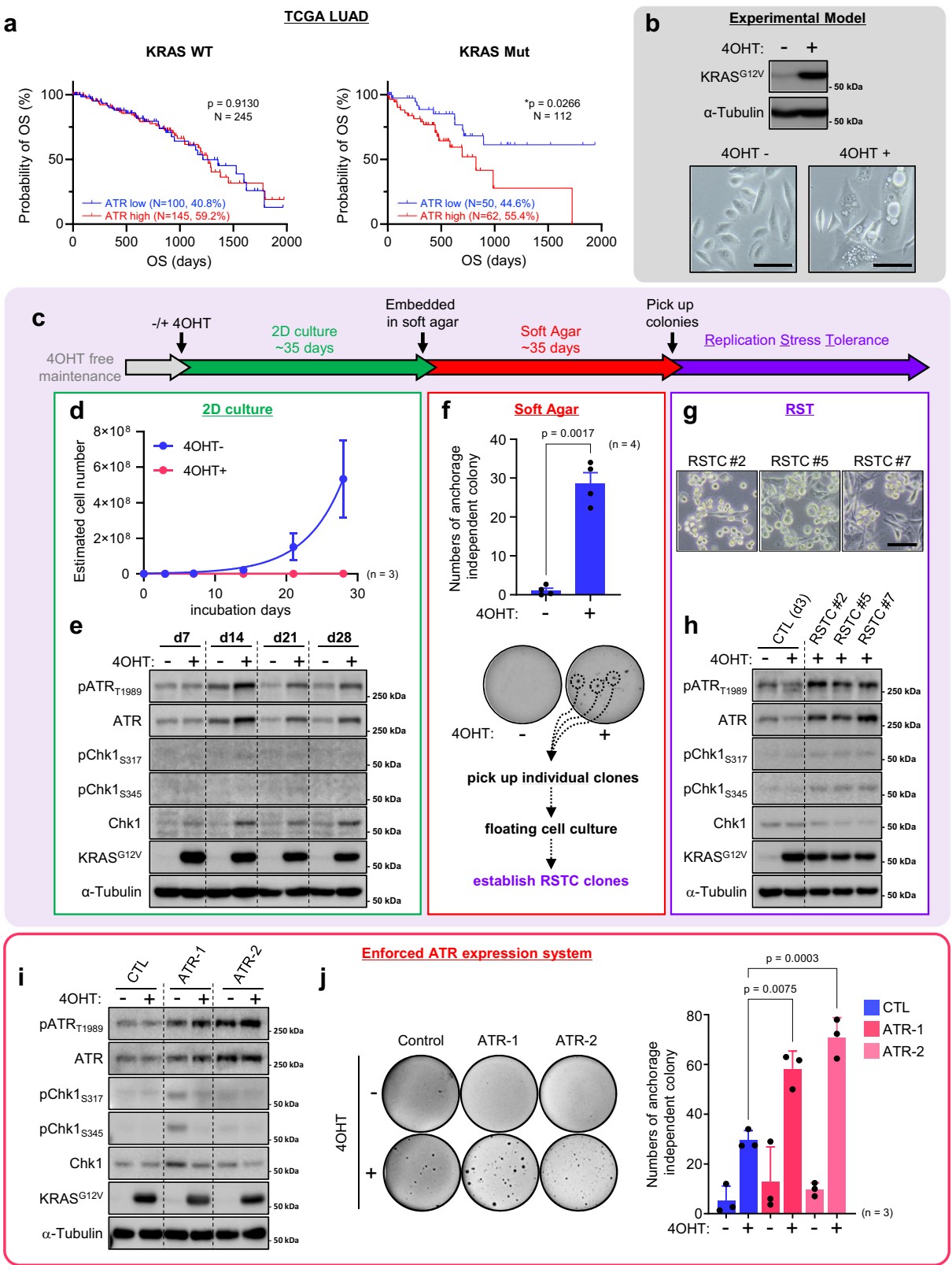

(Supplementary Fig. 1o), suggesting that EMT may enhance cell survival and anchorage-independent growth in the presence of KRAS$^{G12V}$ expression. However, ATR-1 cells without KRAS$^{G12V}$ expression also exhibited the EMT phenotype but did not exhibit enhanced colony formation (Fig. 1j and Supplementary Fig. 1o), suggesting that EMT is not sufficient for the acquisition of anchorage-independent growth. Collectively, these results indicate that elevated expression of ATR in

SAECs is necessary and sufficient to withstand the lethal effects of oncogenic KRAS$^{G12V}$ and to promote RSTC development.

## Elevated ATR expression maintains fork speed by promoting PrimPol-dependent repriming

To reveal how ATR ensures cell survival under exposure to KRAS$^{G12V}$-induced acute and chronic RS, we used a DNA fiber assay[40,41] to

**Fig. 1 | Elevated ATR expression promotes cell survival in the presence of KRAS[G12V] expression. a** High expression of ATR is associated with poor prognosis of lung adenocarcinoma (LUAD) patients. Overall Survival (OS) according to ATR mRNA expression from 357 of LUAD patients harboring KRAS[WT] or KRAS[Mut] were analyzed. Log-rank *p*-values are shown. **b** Top, Control cells harboring estrogen receptor (ER)-KRAS[G12V] were treated with or without 0.1 μM of 4OHT for 3 days. The expression of 4OHT-inducible ER-KRAS[G12V] and α-Tubulin were analyzed by western blotting. Bottom, representative cell image of Control cells with or without (ER)-KRAS[G12V] expression from three independent reproducible experiments. Scale bar = 100 μm. **c** Schematic of the long-term culture of KRAS[G12V]-induced cells. After ~35 days, a few clones adapt to anchorage independent growth culture, acquiring replication stress tolerance ability. **d**, **e** Long term control cell culture with or without 0.1 μM of 4OHT for 28 days. **d** Estimated cell number. The results represent the means ± SEM of three independent experiments. **e** The indicated proteins expression level in control cells maintained with or without 0.1 μM of 4OHT-containing medium for ~28 days were analyzed by western blotting. **f** Control SAECs were grown in 0.4% of soft agar medium with or without 0.1 μM of 4OHT for totally ~70 days. Top, numbers of colony were shown. The results represent the means ± SEM of four independent experiments. two-tailed paired parametric *t*-test. Bottom, representative image of anchorage independent growth assay. Colonies were visualized with Crystal Violet staining. **g**, **h** The characterization of Replication Stress Tolerant Cell (RSTC) clones #2, #5 and #7 maintained in 0.1 μM of 4OHT-containing medium from three independent reproducible experiments. **g** Representative image of RSTC clones. Scale bar = 100 μm. **h** The indicated proteins expression level were analyzed by western blotting. Parental cells treated with or without 0.1 μM of 4OHT for 3 days were shown as control (CTL). **i** Control, ATR-1 and ATR-2 cells were treated with or without 0.1 μM of 4OHT for 3 days. The indicated proteins expression levels were analyzed by western blot analysis. **j** Left, representative image of anchorage independent growth assay of Control, ATR-1 and ATR-2 after 0.1 μM of 4OHT treatment for totally ~70 days. Right, numbers of colony were shown. The results represent the means ± SEM of three independent experiments. one-way ANOVA Tukey's test. All source data are provided as a Source Data file.

monitor DNA replication fork progression (Fig. 2a). After the induction of KRAS[G12V] expression for 3 days, we transiently treated cells with 5-iodo-2′-deoxyuridine (IdU) and 5-chloro-2′-deoxyuridine (CldU) prior to staining the incorporated thymidine analogs with denaturing protocols. KRAS[G12V] expression markedly decreased the DNA fiber length in control cells (Fig. 2a, b)[12]. Since RS can induce the accumulation of replication protein A (RPA)-ssDNA at or behind stalled forks, we analyzed the exposure of ssDNA in the genome by native bromodeoxyuridine (BrdU) staining[42] and found that exposed ssDNA weakly but significantly increased after 3 days of KRAS[G12V] induction (Supplementary Fig. 2a). Accordingly, chromatin-bound RPA32 levels did not increase robustly, but RPA32 phosphorylation on Ser33 increased on day 3 and phosphorylation on Ser4/Ser8 increased on day 7 after KRAS[G12V] induction (Supplementary Fig. 2b, c), while the levels of γH2AX did not increase on day 7 in our experimental conditions (Supplementary Fig. 2d). RPA phosphorylation data suggest that KRAS[G12V] induces RS by slowing or stalling replication forks, but these forks do not collapse synchronously, leaving gH2AX levels largely unchanged. Gradual collapse of stressed forks may lead to cell death over time. In sharp contrast, in ATR-1/−2 cells, increased expression of ATR led to unrestrained fork progression in the presence of KRAS[G12V] expression but did not cause fork speeding under unchallenged conditions (Fig. 2a, b). Unexpectedly, KRAS[G12V] did not affect fork symmetry in either control (with fork slowing) or ATR-1/-2 cells (with unrestrained fork progression) (Supplementary Fig. 2e). To determine whether ATR kinase activity is required for unrestrained fork progression in the presence of KRAS[G12V] expression, an ATR inhibitor (ATRi: berzosertib) was utilized. Because even short-term use of high concentrations of ATRi causes unscheduled origin firing and reduces fork velocity[43,44], a low concentration of an ATRi (1 nM), which does not affect the normal cell cycle and replication fork progression[45], was used in the following assay. Treatment with 1 nM ATRi for 24 h, which did not perturb fork progression in unchallenged ATR-1 cells (Fig. 2c) and did not affect the ATR expression level (Supplementary Fig. 2f), reduced fork velocity in the presence of KRAS[G12V] expression (Fig. 2c), suggesting that ATR kinase activity is required for unrestrained fork progression in the presence of KRAS[G12V] expression in ATR-1 cells. These results indicate that increased ATR protein expression and ATR kinase activity are required for ensuring replication fork progression under KRAS[G12V]-induced acute RS.

Recent studies have shown that DNA damage tolerance (DDT) mechanisms allow cells to prevent replication fork stalling to overcome obstacles during DNA replication[32]. DDT mechanisms broadly include translesion DNA synthesis (TLS), template switching (TS) and replication fork repriming pathways. PrimPol, which possesses both primase and polymerase activities, is emerging as a key player in repriming during the RS response in mammalian cells[27,29]. PrimPol-mediated repriming causes discontinuous replication, leaving unreplicated ssDNA gaps to be filled postreplicatively through either TLS or TS[46]. To investigate whether repriming is involved in the unrestrained DNA replication in ATR-1 cells, we performed a modified DNA fiber assay coupled with S1 nuclease digestion[47]. After IdU and CldU labelling, cells were permeabilized and treated with S1 nuclease. If ssDNA gaps are formed during replication, ssDNA regions are nicked by the S1 nuclease, generating shorter fibers (Fig. 2d). We used 10 U/ml S1 nuclease, a concentration that did not result in DNA fiber cleavage in cells without KRAS[G12V] expression (Fig. 2e). While the already shortened fibers in control cells with KRAS[G12V] expression were not sensitive to S1 treatment, DNA fibers in ATR-1 cells expressing KRAS[G12V], in which replication was unrestrained, were sensitive to S1 nuclease digestion (Fig. 2e). Next, we evaluated ssDNA generation by immunofluorescence analysis of RPA32 binding to ssDNA. The intensity of chromatin-bound RPA32 was similar throughout the cell cycle in ATR-1 cells with or without KRAS[G12V] expression, suggesting that the RPA32 intensity was unable to discriminate repriming-dependent ssDNA gaps because of ssDNA exposed in the replicating genome (Supplementary Fig. 2g). Interestingly, intensity of RPA32 phosphorylated at Ser33 increased significantly when KRAS[G12V] was induced in ATR-1 cells, suggesting a feed-forward loop in which KRAS-dependent stress activates ATR to enable repriming and the ssDNA gap generated by KRAS-induced repriming results in ATR activation (Supplementary Fig. 2g). These results suggest that PrimPol-mediated repriming-dependent ssDNA gaps covered by phosphorylated RPA32 accumulate in KRAS[G12V]-induced ATR-1 cells. Depletion of PrimPol in KRAS[G12V]-induced ATR-1 cells shortened the DNA fibers to a length similar to that in control cells, indicating that PrimPol is required for fork speed maintenance in challenged ATR-1 cells (Fig. 2f and Supplementary Fig. 2h, i). Under this condition, fork stalling or fork reversal may occur in ATR-1 cells with PrimPol depletion and KRAS[G12V] induction, similar to control cells expressing KRAS[G12V] (Fig. 2f). Importantly, these shortened fibers are no longer sensitive to S1 nuclease treatment, demonstrating that PrimPol is responsible for ssDNA gap accumulation in response to KRAS[G12V] expression in ATR-1 cells (Fig. 2f). Similar observations—i.e., PrimPol-dependent unrestrained fork progression under KRAS[G12V]-induced RS in cells with increased ATR expression—were made in another human normal epithelial cell line, retinal pigment epithelium (RPE) cells, which were immortalized by introducing h-TERT (Supplementary Figs. 2h and 2j−m). This finding suggests that these phenotypes did not result from transduction of the CDK4 mutant and CyclinD1 in SAECs.

Next, we analyzed replication fork progression in RSTCs that adapted to chronic KRAS[G12V]-induced RS (Fig. 1c−h). Although KRAS[G12V] was sustainably expressed (Fig. 1h), RSTCs showed unrestrained fork

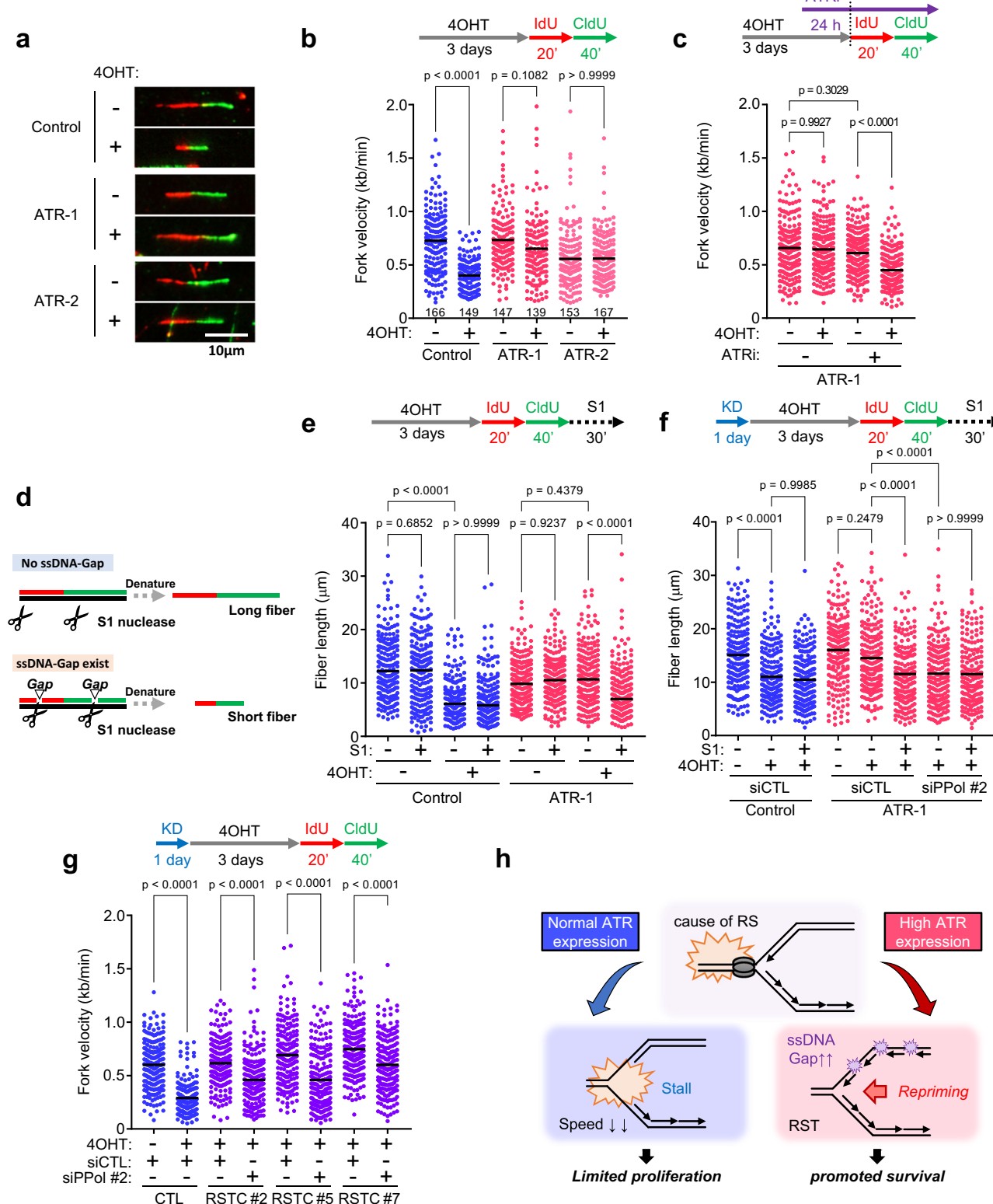

progression with no effect on fork symmetry (Fig. 2g and Supplementary Fig. 2n), similar to ATR-1 cells with KRAS^G12V expression (Fig. 2b). Notably, PrimPol mRNA expression was slightly increased in all RSTCs (Supplementary Fig. 2o), and depletion of PrimPol reduced the fork speed in RSTCs, suggesting that PrimPol is required for fork speed maintenance after tolerance acquisition under chronic RS adaptation (Fig. 2g and Supplementary Fig. 2p). These results strongly suggest that ATR promotes PrimPol-dependent repriming to maintain

fork speed, and, thereby, cellular tolerance to KRAS^G12V-induced acute and chronic RS (Fig. 2h).

**PrimPol-dependent repriming requires ATR-Chk1 kinase activity**
To assess whether PrimPol activity is responsible for repriming in ATR-1 cells, we introduced mutations in the C-terminal zinc finger domain of PrimPol (C419G/H426Y, termed the CH mutant), thus disrupting its primase activity[28], and established cells with doxycycline-inducible

**Fig. 2 | Elevated ATR expression maintains fork speed by promoting PrimPol-dependent repriming. a**, **b** Fork speed in Control, ATR-1 and ATR-2 cells treated with 0.1 μM of 4OHT for 3 days. **a** Representative images of DNA fibers. IdU (red) was treated for 20 min followed by CldU (green) for 40 min, then visualized by denature protocol. Scale bar = 10 μm. **b** Dot plot and mean of fork speed. Representative result of three independent reproducible experiments are shown. Black lines indicate the mean; $n \geq 139$; one-way ANOVA Tukey's test. **c** Dot plot and mean of fork speed in ATR-1 cells treated with 0.1 μM of 4OHT for 3 days. 1 nM of ATRi (Berzosertib) was added 24 h prior to IdU/CldU labeling. Representative result of two independent reproducible experiments are shown. **d** Schematic for detection of ssDNA gap by ssDNA specific S1 nuclease. Before denaturing of DNA fiber, DNA were cleaved by 10 U/ml of S1 nuclease. **e** Dot plot and mean of fiber lengths in Control and ATR-1 cells treated with 0.1 μM of 4OHT for 3 days with or without 10 U/ml of S1 nuclease for

30 min. Representative result of three independent reproducible experiments are shown. **f** Dot plot and mean of fiber lengths in Control and ATR-1 cells transfected with 1 nM of siControl or siPrimPol for 24 h and treated with 0.1 μM of 4OHT for 3 days with or without 10 U/ml of S1 nuclease for 30 min. Representative result of two independent reproducible experiments are shown. **g** Dot plot and mean of fork speed in parental Control cells and RSTC clones #2, #5 and #7 transfected with 1 nM of siControl or siPrimPol for 24 h and treated with 0.1 μM of 4OHT for 3 days. Representative result of two independent reproducible experiments are shown. **h** Proposed ATR-PrimPol mediated RST model. In response to acute and chronic RS induced by $KRAS^{G12V}$, elevated ATR expression promotes PrimPol-dependent repriming to maintain fork speed but allows cell to generate ssDNA, resulting in high risk of genomic instability. **c, e, f, g** Black lines indicate the mean; $n = 200$; one-way ANOVA Tukey's test. All source data are provided as a Source Data file.

expression of myc-PrimPol^WT or myc-PrimPol^CH (Fig. 3a). In ATR cells with depletion of endo-PrimPol by siRNA targeting the UTR of PrimPol (siPPol#4) (Supplementary Fig. 3a), unrestrained fork progression was inhibited, confirming previous results (Figs. 3b, 2f, and Supplementary Fig. 2i). Upon induction of myc-PrimPol^WT in PrimPol-depleted ATR-1 cells, the reduction in fork speed under $KRAS^{G12V}$-induced RS was clearly rescued, but this effect was not seen in cells expressing the myc-PrimPol^CH mutant, suggesting that primase activity of PrimPol is required for repriming in ATR-1 cells (Fig. 3b). Interestingly, even in control SAECs, myc-PrimPol^WT expression was sufficient to induce unrestrained fork progression under $KRAS^{G12V}$-induced RS in an ATR kinase-dependent manner (Fig. 3c). ATRi treatment did not affect the expression level or chromatin loading of myc-PrimPol^WT upon $KRAS^{G12V}$ expression, suggesting that ATR kinase activity is critical for triggering PrimPol-mediated repriming on top of PrimPol expression level (Supplementary Fig. 3b, c)[46]. Moreover, myc-PrimPol^WT did not cause fork speeding in control and ATR-1 cells without $KRAS^{G12V}$ expression, suggesting that PrimPol-mediated repriming is carried out only in the presence of RS (Fig. 3b, c).

Next, to evaluate the possibility that the ATR kinase cascade is involved in PrimPol regulation, we analyzed the phosphorylation of Chk1 at Ser345, representing activated Chk1. While Chk1 Ser345 phosphorylation was detected only on day 1 and was almost completely abolished after day 2 in control SAECs, it was detected until day 2 in ATR-1 cells (Supplementary Fig. 3d), suggesting that prolonged activation of Chk1 may be involved in the initiation of repriming in response to $KRAS^{G12V}$-induced RS. Consequently, short-term treatment with an ATRi at a high concentration (1 μM), which caused a reduced fork speed in both unperturbed control and ATR-1 cells, was not able to suppress unrestrained fork progression mediated by PrimPol-dependent repriming (Supplementary Fig. 3e). Furthermore, treatment with 1 nM Chk1 inhibitor (Chk1i: rabusertib) suppressed unrestrained fork progression to a similar level as 1 nM ATRi treatment in ATR-1 cells expressing $KRAS^{G12V}$ (Fig. 3d). In RSTCs, these inhibitors also suppressed fork progression (Fig. 3e) to the same extent as that in cells depleted of PrimPol (Fig. 2g and Supplementary Fig. 2p). We also analyzed the phosphorylation of Chk1, but under the same condition, these inhibitors did not alter Chk1 phosphorylation levels at Ser345 or Ser317 in RSTCs (Supplementary Fig. 3f), suggesting that even very weak ATR and Chk1 inhibition with the respective inhibitors was sufficient to suppress PrimPol-dependent repriming. (Fig. 3e and Supplementary Fig. 3f). Importantly, simultaneous treatment with the ATRi and Chk1i did not further suppress fork progression, indicating that ATR and Chk1 regulate PrimPol-mediated repriming in an epistatic manner (Fig. 3d, e). Moreover, both the ATRi and Chk1i inhibited RSTC colony reformation in a dose dependent manner, suggesting that the activity of both ATR and Chk1 is required for maintaining fork progression and anchorage-independent growth in RSTCs (Fig. 3f). Next, to determine whether PrimPol is phosphorylated during activation, we used mass spectrometry to analyze myc-PrimPol^WT purified from ATR-1 cells expressing $KRAS^{G12V}$ (Supplementary Fig. 3g). PrimPol was

phosphorylated at Ser26, Ser33, and Ser255 and dephosphorylated at Ser207 and possibly at Ser489 and Ser499 (Supplementary Fig. 3h) upon $KRAS^{G12V}$-induced RS. Phosphorylation of Ser255 was previously shown by others (PhoshoSitePlus®: https://www.phosphosite.org)[33]. The location of Ser255 in the unstructured domain spanning archaeal eukaryotic primase (AEP) motifs II and III, a potential regulatory element conserved in PrimPol among higher primates, prompted us to focus on this phosphorylation site (Fig. 3g). We next generated a nonphosphorylatable S255A mutant and a phosphomimetic S255D mutant to investigate whether Ser255 phosphorylation is implicated in PrimPol activation (Fig. 3h). Unlike PrimPol^WT, PrimPol^S255A was unable to rescue the reduction in fork speed under $KRAS^{G12V}$-induced RS, whereas PrimPol^S255D did so efficiently (Fig. 3i). Furthermore, the PrimPol^S255D-mediated unrestrained fork progression was no longer ATRi-sensitive, suggesting that phosphorylation of PrimPol at Ser255 is crucial for its repriming activity. Thus, $KRAS^{G12V}$-induced RS induces phosphorylation of PrimPol at Ser255 and promotes repriming to maintain fork progression in an ATR/Chk1-dependent manner.

## KRAS^G12V-induced transcription-dependent chromatin remodeling causes RS

Next, we sought to determine the cause of $KRAS^{G12V}$-induced RS. The expression of oncogenic RAS has been linked to RS due to an increase in global transcription activity[12]. Based on these findings, we tested whether 5,6-dichloro-1-β-D-ribofuranosylbenzimidazole (DRB), an inhibitor of CDK9 required for RNA polymerase II regulation, can reverse the reduction in fork progression. A short pulse treatment of DRB was sufficient to restore fork progression (Fig. 4a). In addition, nascent RNA synthesis was quantified by assessing nuclear incorporation of the modified RNA precursor 5-ethynyluridine (EU) for 1 h. $KRAS^{G12V}$ induction was found to slightly increase transcriptional activity in our experimental model of control SAECs and ATR-1 cells, and DRB treatment greatly reduced RNA synthesis (Fig. 4b), suggesting that inhibition of RNA transcription is sufficient to reverse the reduction in fork progression. In addition, the global transcription level was not decreased in ATR-1 cells, suggesting that unrestrained fork progression in ATR-1 cells does not result from reduced transcription (Figs. 2b and 4a, b). We next sought to verify whether R-loops, which can act as obstacles to replication fork progression, accumulate in cells expressing $KRAS^{G12V}$. Slot blot analysis of total DNA with the S9.6 monoclonal antibody, which recognizes DNA:RNA hybrids, confirmed that R-loop levels were not increased by $KRAS^{G12V}$ expression in control SAECs and ATR-1 cells (Supplementary Fig. 4a). Moreover, the reduction in fork progression was not reversed in SAECs expressing RNaseH1, although the R-loop level was slightly decreased (Supplementary Fig. 4b–d). The S9.6 signal was reduced by lentiviral transduction of RNaseH1 in SAECs (Supplementary Fig. 4b) and disappeared when the extracted DNA was digested with RNaseH1 (exRNH), confirming that our experiment was well controlled (Supplementary Fig. 4a, b). We also analyzed transcription-replication collision (TRC), another obstacle to replication fork progression caused by increased transcriptional

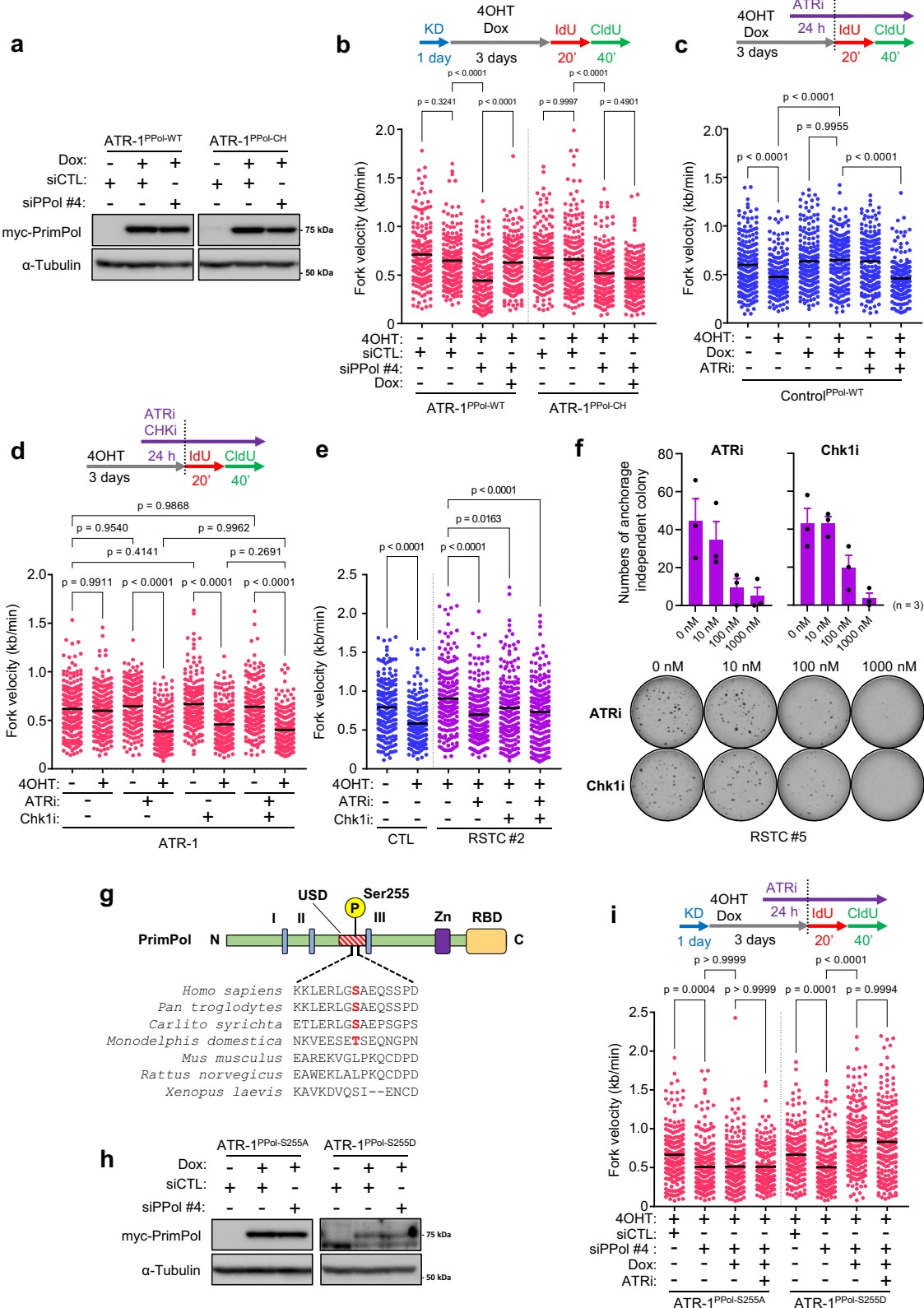

activity[48], and found that TRC was not increased in control SAECs or ATR-1 cells expressing KRAS[G12V] (Supplementary Fig. 4e). Furthermore, DRB, which reversed the reduction in fork progression, did not significantly decrease the R-loop level or TRCs. Collectively, these results indicate that although RNA transcription causes reduced replication fork progression, the resulting R-loops and TRCs are not critical determinants of the reduction in fork progression induced by

KRAS[G12V], and the increased ATR expression does not resolve RNA-related impediments.

To further determine how RNA transcription is involved in the cause of KRAS[G12V]-induced RS, we investigated whether RNA transcription stabilizes total RNA by labeling cells with EU for 23 h and then releasing them into EU-free medium in the presence of DRB for 1 h before the EU-click reaction. DRB treatment significantly reduced the

**Fig. 3 | PrimPol-dependent repriming requires ATR-CHK1 kinase activity.**
**a, b** ATR-1 cells harboring doxycycline-inducible myc-PrimPol[WT] or myc-PrimPol[CH] were transfected with 5 nM of siControl or siUTR-PrimPol (siPPol #4) for 24 h and treated with 1 µg/ml of doxycycline for 24 h. The expression level of myc-tag PrimPol and α-Tubulin were analyzed by western blotting. **b** Dot plot and mean of fork speed in ATR-1 cells harboring doxycycline-inducible myc-PrimPol[WT] or myc-PrimPol[CH] transfected with 5 nM of siControl or siPPol #4 for 24 h and treated with 0.1 µM of 4OHT with or without 1 µg/ml of doxycycline for 3 days. Representative result of two independent reproducible experiments are shown. **c** Dot plot and mean of fork speed in control cells harboring doxycycline-inducible myc-PrimPol[WT] treated with 0.1 µM of 4OHT with or without 1 µg/ml of doxycycline for 3 days. 1 nM of ATRi (Berzosertib) was added 24 h prior to IdU/CldU labeling. Representative result of two independent reproducible experiments are shown. **d** Dot plot and mean of fork speed in ATR-1 cells treated with 0.1 µM of 4OHT for 3 days. Low dose of ATRi (1 nM) and Chk1i (Rabusertib, 1 nM) was added 24 h prior to IdU/CldU labeling. Representative result of two independent reproducible experiments are shown. **e** Dot plot and mean of fork speed in Control and RSTC #2 cells treated with 0.1 µM of 4OHT for 3 days. Low dose of ATRi (1 nM) and Chk1i (1 nM) was added 24 h prior to IdU/CldU labeling. Representative result of two independent reproducible

experiments are shown. **f** Colony re-formation assay of RSTC #5 with long-term ATR or Chk1 inhibition. Top, the number of colonies treated at indicated concentration of ATR or Chk1 inhibitor for ~14 days. The results represent the means ± SEM of three independent experiments. Bottom, representative image of anchorage independent colonies. **g** Protein domain structure of PrimPol. The catalytic signature motifs (I, II, and III) of archaeal-eukaryotic primase (AEP) domain; Zinc-finger domain (Zn) and RPA binding domain (RBD) are indicated. Multiple sequence alignment of PrimPol homologs in indicated animals is shown. Phosphorylation sites (S/T) are indicated in red. **h** ATR-1 cells harboring doxycycline-inducible myc-PrimPol[S255A] or myc-PrimPol[S255D] were transfected with 5 nM of siControl or siPPol #4 for 24 h and treated with 1 µg/ml of doxycycline for 24 h. The expression level of myc-tag PrimPol and α-Tubulin were analyzed by western blotting. **i** Dot plot and mean of fork speed in ATR-1 cells harboring doxycycline-inducible myc-PrimPol[S255A] or myc-PrimPol[S255D] transfected with 5 nM of siControl or siPPol #4 for 24 h and treated with 0.1 µM of 4OHT with or without 1 µg/ml of doxycycline for 3 days. 1 nM of ATRi was added 24 h prior to IdU/CldU labeling. Representative result of four independent reproducible experiments are shown. **b, c, d, e, i** Black lines indicate the mean; *n* = 200; one-way ANOVA Tukey's test. All source data are provided as a Source Data file.

---

total RNA level in control cells expressing KRAS[G12V], and a similar trend was observed in ATR-1 cells expressing KRAS[G12V], suggesting that RNA transcription is required for stabilizing total RNA in KRAS[G12V]-induced cells (Fig. 4c and Supplementary Fig. 4f). Interestingly, RNA sequencing (RNA-seq) analysis revealed that the genes whose expression was reduced by DRB treatment in KRAS[G12V]-induced cells showed a higher H3K27me3 signal in chromatin immunoprecipitation sequencing (ChIP-seq) analysis of control SAECs (Supplementary Fig. 4g, h)[49]. Therefore, we hypothesized that KRAS[G12V]-induced RNA transcription recruits Polycomb repressive complex 2 (PRC2) complexes to chromatin, where it deposits the H3K27me3 mark of facultative heterochromatin[50]. Accordingly, expression of KRAS[G12V] increased the levels of H3K27me3 (Fig. 4d, e) and, simultaneously, H3K9me3 (Supplementary Fig. 4i). However, H3K27me3 but not H3K9me3 was inhibited by DRB, suggesting that chromatin remodeling via H3K27me3 is involved in RS induced by KRAS[G12V] expression (Fig. 4d, e and Supplementary Fig. 4i). Consistent with our hypothesis, an Enhancer of zeste 2 (EZH2) inhibitor (GSK126), which inhibited H3K27me3 (Fig. 4d, e), reversed the reduction in fork progression induced by KRAS[G12V] (Fig. 4f). These results suggest that KRAS[G12V] induces the RNA transcription from H3K27me3-enriched genes, which stabilizes total RNA that likely promotes the formation of spatial compartments[51], thus recruiting PRC2 complexes to chromatin[50] for further methylation on Histone H3.

According to recent studies, chromatin structures present in heterochromatin hinder replication fork progression and cause RS[52,53]. Interestingly, chloroquine (CQ), a DNA-intercalating drug[54] and an epigenetic modulator of H3K27me3[55] that relaxes chromatin, reversed the reduction in fork progression but did not inhibit nascent RNA synthesis (Fig. 4a, b), and both DRB and CQ restored fork progression in PrimPol-depleted ATR-1 cells (Supplementary Fig. 4j). Furthermore, we performed a micrococcal nuclease (MNase) sensitivity assay to address chromatin condensation status[56]. Control SAECs expressing KRAS[G12V] treated with DRB, CQ, or GSK126 exhibited an increase in MNase accessibility of their chromatin compared to those treated with DMSO suggesting that KRAS[G12V]-induced chromatin compaction is indeed decondensed by these drugs (Fig. 4g). These results suggest that chromatin compaction associated with KRAS[G12V]-mediated transcription-dependent increases in H3K27me3 may cause RS in both control SAECs and ATR-1 cells (Fig. 4h).

## PrimPol-dependent repriming occurs at locally compacted chromatin regions

Although H3K27me3 and H3K9me3 are generally defined as hallmarks of facultative and constitutive heterochromatin, respectively, it

recently became evident that these marks frequently colocalize in the genome[57]. Heterochromatin protein 1 (HP1) is a key structural adapter required for heterochromatin formation and maintenance, and its binding to H3K9me3 is enhanced in the presence of H3K27me3[58]. To verify whether the HP1 interaction is involved in the cause of KRAS[G12V]-induced RS, HP1α and HP1β were depleted (Fig. 5a). Interestingly, depletion of either HP1α or HP1β completely rescued the restrained fork progression in control SAECs expressing KRAS[G12V] (Fig. 5b and Supplementary Fig. 5a) and reintroduction of HP1α reproduced the suppressed fork progression (Fig. 5c, d), indicating that HP1 forms an obstacle for replication. Moreover, HP1α depletion suppressed chromatin-bound RPA induced by KRAS[G12V] expression (Supplementary Fig. 5b) and partially rescued the cell growth inhibition caused by KRAS[G12V] expression (Supplementary Fig. 5c), suggesting that depletion of HP1α suppressed the RS in cells expressing KRAS[G12V], allowing them to continue replication and thus restoring cell growth. Together, our results suggest that both H3K27me3 and HP1 contribute to the KRAS[G12V]-induced RS. While it is conceivable that H3K27me3 promotes the binding of HP1 to preexisting H3K9me3 in this context, it is also possible that undetectable changes in H3K9me3 allow HP1 to act in parallel with H3K27me3 to establish heterochromatin and impede replication forks.

Subsequently, we evaluated whether PrimPol conducts repriming near locally compacted chromatin regions. To detect repriming-dependent ssDNA gap exposure on the parental DNA, ATR-1 cells were grown in 5-bromo-2'-deoxyuridine (BrdU)-containing medium for the first 48 h during KRAS[G12V] expression before being released into BrdU-free medium for 24 h[59] and 5-ethynyl-2'-deoxyuridine (EdU)-containing medium for 30 min to label the S-phase cells. We then performed a proximity ligation assay (PLA) using antibodies against H3K27me3 and BrdU under nondenaturing conditions (Fig. 5e). PLA foci were detected only in cells cultured in BrdU-containing medium and were juxtaposed but not colocalized with nascent DNA (EdU), validating the accuracy of our PLA for monitoring the proximity of incorporated BrdU and H3K27me3 (Fig. 5f). Remarkably, KRAS[G12V] expression increased the number of PLA foci in S-phase cells (Fig. 5g, h) in a PrimPol expression-dependent manner (Fig. 5i, and Supplementary Fig. 5d, e), suggesting that PrimPol-mediated repriming generates ssDNA gaps near locally compacted chromatin regions containing H3K27me3. Interestingly, the increase in PLA foci was observed in not only S-phase but also G2/M-phase cells (Fig. 5g, h), suggesting that repriming-dependent ssDNA gaps generated in S-phase, when DNA is replicated, may persist into G2/M phase, resulting in genomic instability in ATR-1 cells.

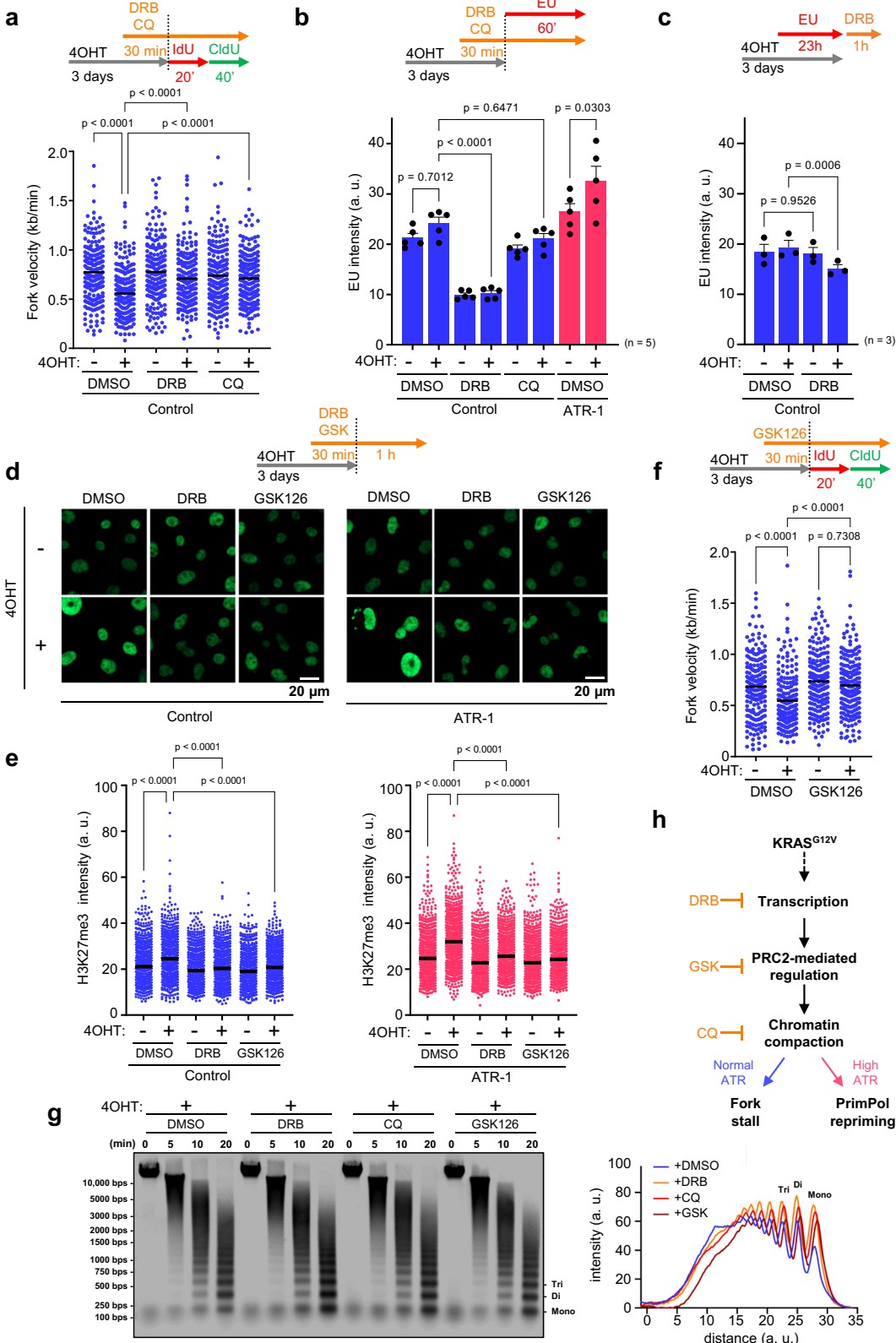

## Genomic instability driven by KRAS^G12V is tolerated in cells with elevated ATR expression

To determine the impact of increased ATR expression under KRAS$^{G12V}$-induced RS on genomic instability, we examined genomic instability markers, including micronuclei (MN), and blebs[60] (Fig. 6a). Under unchallenged conditions, approximately 5% of control SAECs were positive for MN, and KRAS$^{G12V}$ expression tended to increase the percentage to ~10% (Fig. 6b). Remarkably, MN formation was significantly promoted in ATR-1 cells expressing KRAS$^{G12V}$ compared to control SAECs, and similar trends were observed for bleb formation (Fig. 6b). Anaphase bridges in M phase cells, consequences of RS[61], were also slightly increased by KRAS$^{G12V}$ expression in both control SAECs and ATR-1 cells, and ATR-1 cells expressing KRAS$^{G12V}$ contained ~1.5-fold more anaphase bridge-positive cells than control SAECs

**Fig. 4 | KRAS$^{G12V}$-induced transcription-dependent chromatin remodeling causes RS. a** Dot plot and mean of fork speed in Control cells treated with 0.1 μM of 4OHT for 3 days. 100 μM of DRB or 40 μM of Chloroquine (CQ) was added 30 min prior to IdU/CldU labeling. Representative result of three independent reproducible experiments are shown. **b** Quantification of EU intensity of Control and ATR-1 cells treated with 0.1 μM of 4OHT for 3 days. 100 μM of DRB or 40 μM of CQ was added 30 min prior to 1 mM of EU labeling for final 60 min. The results represent the means ± SEM of five independent experiments. one-way ANOVA Tukey's test. arbitrary units, a. u. **c** Quantification of total RNA of Control cells treated with 0.1 μM of 4OHT for 3 days. 1 mM of EU was added 23 h prior to 100 μM of DRB treatment for final 60 min. The results represent the means ± SEM of three independent experiments. one-way ANOVA Tukey's test. arbitrary units, a. u. **d**, **e** After 0.1 μM of 4OHT treatment for 3 days, Control and ATR-1 cells were treated with 100 μM of DRB or 2.5 μM of GSK126 for 90 min, followed by staining with anti-H3K27me3 antibody with pre-extraction method. **d** Representative image of chromatin-bound H3K27me3 staining. Scale bar = 20 μm. **e** Quantification of the H3K27me3 intensity shown in (**d**). Representative result of three independent

reproducible experiments are shown. Black lines indicate the mean; arbitrary units, a. u.; n = 1000; one-way ANOVA Tukey's test. **f** Dot plot and mean of fork speed in Control cells treated with 0.1 μM of 4OHT for 3 days with or without 2.5 μM of GSK126. GSK126 was added 30 min prior to IdU/CldU labeling. Representative result of four independent reproducible experiments are shown. **g** Left, representative result of MNase sensitivity assay. After 0.1 μM of 4OHT treatment for 3 days, Control cells treated with 100 μM of DRB, 2.5 μM of GSK126 or 40 μM of CQ for 90 min, followed by permeabilization and MNase digestion for 5, 10, 20 min respectively. Right, quantification of digested DNA intensity. Representative result of two independent reproducible experiments are shown. arbitrary units, a. u. **h** The model of KRAS$^{G12V}$-induced RS and its response. KRAS$^{G12V}$ induces the transcription of nascent RNA enriched in polycomb repressive complex 2 (PRC2)-regulated genes leading to PRC2 recruitment and trimethylation of H3K27, generating locally compacted heterochromatin region, resulting in a cause of RS. **a**, **f** Black lines indicate the mean; n = 200; one-way ANOVA Tukey's test. All source data are provided as a Source Data file.

(Supplementary Fig. 6a). In control cells, KRAS$^{G12V}$ expression reduced cell survival to ~50%, accompanied by methuosis-like cell death (Fig. 1b), and ATRi treatment further decreased cell survival (Fig. 6c), suggesting that KRAS$^{G12V}$-induced stalled forks may gradually collapse after ATR inhibition, causing cell death. In ATR-1 cells, KRAS$^{G12V}$ expression did not decrease cell survival following increased MN formation, while inhibition of ATR reduced cell survival and MN formation (Fig. 6c, d). These findings suggested that ATR inhibition was unable to induce PrimPol-dependent repriming for tolerance to KRAS$^{G12V}$-induced RS (Fig. 3d); therefore, the stalled forks potentially collapsed, leading to cell death. However, how specific PrimPol-mediated repriming contributes to MN formation in ATR-1 cells with KRAS$^{G12V}$ expression and their survival still needs further investigation. These results suggest that ATR kinase activity is required for cell survival, allowing enhanced genomic instability during KRAS$^{G12V}$-induced RS. Notably, ATR-1 cells without KRAS$^{G12V}$ expression tended to exhibit enhanced nuclear abnormalities, suggesting that elevated ATR expression alone poses a risk of genomic instability in response to endogenous RS (Fig. 6b). Consistent with the observations of ATR-1 cells, RSTCs derived from control SAECs, in which the elevated ATR level was maintained, also showed increased MN formation (Fig. 6e). WGS analysis of RSTCs revealed an increase in the total number of structural variants (SVs), most of which were deletions (Fig. 6f), frequent copy number variations (CNVs), and genomic rearrangements (Fig. 6g). We also found that all RSTC clones exhibited whole-genome duplication (WGD)[62], which is consistent with a recent study showing that oncogenes can drive WGD through RS[63]. These findings suggest that the ATR-PrimPol-mediated tolerance mechanism against KRAS$^{G12V}$-induced RS might promote WGD, which is a frequent event in cancer evolution and an important driver of aneuploidy. In addition, RSTCs exhibited a more than twofold increase in single-nucleotide variations (SNVs) (Fig. 6h). We analyzed mutation signatures to determine the mechanism underlying mutational processes and identified COSMIC signature SBS8 only in RSTCs (Supplementary Fig. 6b), which probably arises due to uncorrected late replication errors[64], suggesting that KRAS$^{G12V}$-induced RS drives not only SVs but also mutational processes. Collectively, these results indicate that elevated ATR expression enables cells to survive and accumulate genomic instability under KRAS$^{G12V}$-induced RS, resulting in clonal expansion of RSTCs.

Next, we tested whether ATR-PrimPol-mediated RST is maintained in lung cancer cells with oncogenic *KRAS* mutation (Supplementary Fig. 7a). The *KRAS$^{G12mut}$* cell lines showed significantly higher levels of PrimPol and H3K27me3 than the *KRAS$^{WT}$* cell lines, and the expression levels of ATR and HP1α differed from cell to cell, with a trend toward higher expression in the *KRAS$^{G12mut}$* cell lines (Fig. 7a). Interestingly, depletion of PrimPol in *KRAS$^{G12mut}$* cells reduced the length of DNA

fibers to nearly half that in control cells but had little effect in *KRAS$^{WT}$* cells (Fig. 7b and Supplementary Fig. 7b). These results suggest that *KRAS$^{G12mut}$* cancer cells prone to induce heterochromatin-associated RS and their DNA replication is strongly dependent on PrimPol-mediated repriming consistent with that in RSTCs (Fig. 2g).

Given that ATR promotes RST mediated by PrimPol-dependent repriming to maintain fork speed and ensure cell survival with acquisition of genomic instability in response to KRAS$^{G12V}$-induced RS, we examined whether combined overexpression of PrimPol and ATR could impact OS in KRAS-driven cancer in the same cohort of LUAD patients (shown in Fig. 1a). Patients with higher tumor mRNA levels of both ATR and PrimPol had a lower OS rate than those with *KRAS*-mutant tumors with lower mRNA levels of both ATR and PrimPol among patients with *KRAS*-mutant tumors in the LUAD cohort (Fig. 7c). However, this pattern was not observed for COAD and PAAD (Supplementary Fig. 7c), suggesting that the impact of ATR-PrimPol-mediated RST on OS is more prominent in LUAD than in these other tumor types. To evaluate the observation that the association of high ATR and PrimPol mRNA expression with poor prognosis is an exclusive feature of the TCGA LUAD cohort, we analyzed an additional LUAD cohort from National Cancer Center Japan (NCCJ). To align the patient backgrounds in the TCGA and NCCJ data, we selected smokers and analyzed this group of patients. Patients with higher ATR mRNA expression and patients with higher ATR and PrimPol mRNA expression had significantly lower OS rates than their counterparts with lower expression in both the TCGA and NCCJ cohorts (Supplementary Fig. 7d, e). These results strongly support our observation in SAECs that the higher the level of ATR expression, the more tolerant are cells to KRAS$^{G12V}$-induced RS, and this effect is mediated through promotion of PrimPol-dependent repriming, resulting in an increased risk of genomic instability.

## Discussion

Our results, summarized in Fig. 7d, describe a crucial role of ATR in RST and the survival of lung epithelial cells with oncogenic KRAS expression. KRAS$^{G12V}$ expression induces local genome compaction, the cause of RS, constituted by HP1 binding via H3K27me3, which is methylated in a transcription-dependent manner. Our results show that the ATR-PrimPol pathway functions as a regulatory module at replication forks to complete replication with ssDNA gaps by promoting repriming, allowing cells to survive under KRAS$^{G12V}$-induced RS with acquisition of genomic instability. These findings may underlie the poor prognosis of patients with *KRAS$^{mut}$* lung tumors with high ATR and PrimPol mRNA expression.

How cancer cells develop or emerge has long been one of the most heavily debated questions. Activated oncogenes cause RS and DNA damage in cultured cells, thereby triggering the DNA damage

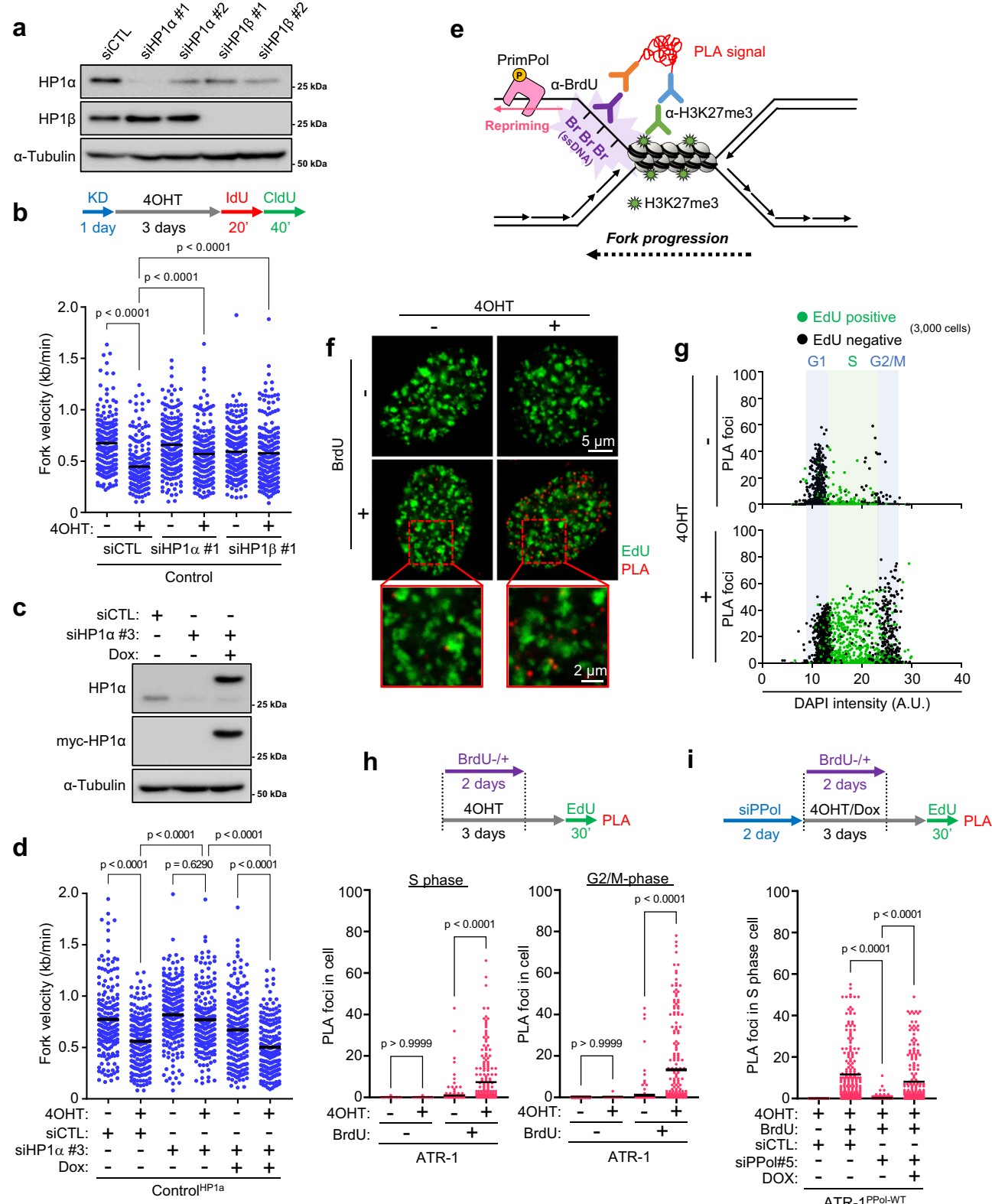

response (DDR) and cell cycle checkpoints that provide a biological anticancer barrier in the early stages of tumorigenesis[65,66]. At the same time, a subset of cells, which might be the origin cells of tumor development, is selected[5]. Previous studies have reported that ATR differentially impacts tumorigenesis depending on its expression level[67]. Although heterozygotes for the *ATR* gene are known to be tumor prone because of a partial reduction in the RS response[20,23], the grossly reduced ATR levels due to a hypomorphic mutation of *ATR* that

mimics Seckel syndrome seems to eliminate cells experiencing suprathreshold RS in the oncogene-transformed cells[24]. Our results suggest that upregulated ATR expression is a critical determinant of survival under KRAS[G12V]-induced RS, adding an unanticipated dosage-dependent function of ATR to the tumor initiation step. Consistent with this idea, supraphysiological levels of ATR module components such as Chk1[25] and Claspin/Timeless[68] promote malignant transformation induced by oncogenic Ras, suggesting that the ATR signaling

**Fig. 5 | PrimPol-dependent repriming occurs at the locally compacted chromatin region. a** Control cells were transfected with 1 nM of two independent siR-NAs of HP1α and HP1β for 24 h. The expression level of HP1α and HP1β were analyzed by western blotting. **b** Dot plot and mean of fork speed in Control cells transfected with 1 nM of siControl, siHP1α or siHP1β for 24 h and treated with 0.1 μM of 4OHT for 3 days. Representative result of two independent reproducible experiments are shown. **c** Control cells harboring doxycycline-inducible myc-HP1α were transfected with 1 nM of siControl or siUTR-HP1α (siHP1α #3) for 24 h and treated with 1 μg/ml of doxycycline for 24 h. The expression level of myc-tag HP1α and α-Tubulin were analyzed by western blotting. **d** Dot plot and mean of fork speed in Control cells harboring doxycycline-inducible myc-HP1α transfected with 1 nM of siControl or siHP1α #3 for 24 h and treated with 0.1 μM of 4OHT with or without 1 μg/ml of doxycycline for 3 days. Representative result of two independent reproducible experiments are shown. **e** Strategy of proximity ligation assay (PLA) to monitor ssDNA exposure near heterochromatin mediated by H3K27me3. After the siRNA transfection, cells were grown in 10 μM of BrdU-containing medium

and incubated for 2 days before released into BrdU-free medium for 1 day. Cells were treated with 10 μM of EdU for final 30 min to label the S-phase cells, then PLA was performed using antibody against BrdU and H3K27me3. **f–h** The result of H3K27me3-BrdU PLA from three independent reproducible experiments with ATR-1 cells treated with 0.1 μM of 4OHT for 3 days. **f** Representative PLA foci image. Scale bar = 5 μm. Expanded images are shown in red square. Scale bar = 2 μm. **g** Scatterplots showing the number of PLA foci. 3,000 cells from each sample were randomly selected and plotted. EdU negative (black) and EdU positive (green) are indicated respectively. **h** Number of PLA foci in S phase or G2/M phase cells defined by EdU incorporation level shown in (**5 g**). **i** Number of PLA foci in S phase cells of ATR-1 cells harboring doxycycline-inducible myc-PrimPol^WT transfected with 1 nM of siControl or siUTR-PrimPol (siPPil #5) for 2 days and treated with 0.1 μM of 4OHT with or without 1 μg/ml of doxycycline for 3 days. Representative result of three independent reproducible experiments are shown. **b, d, h, i** Black lines indicate the mean; *n* = 200; one-way ANOVA Tukey's test. All source data are provided as a Source Data file.

pathway-mediated RS response may help cells to cope with RS and survive despite the enhanced genotoxic stress.

RSTs are involved in the mechanism endowing replication fork plasticity, among which repriming is emerging as a central mechanism. Repriming is activated upon treatment with exogenous damaging agents such as cisplatin, UV-C, and low doses of hydroxyurea (HU), suggesting that repriming is a general mechanism to cope with RS even when replication forks do not encounter DNA lesions[46]. A recent report showed that the expression level and repriming activity of PrimPol are controlled in an ATR-dependent manner[45]. In addition, USP36 participates in PrimPol protein stability by removing Lys29-linked polyubiquitin chains from PrimPol to play a critical role in RST[69]. These findings highlight the importance of the transcript expression level or protein stability of PrimPol for initiating repriming. During KRAS^G12V-induced RS, forced PrimPol expression is sufficient to activate repriming, but the repriming is ATR kinase activity dependent and requires prolonged ATR-Chk1 activation, suggesting that ATR kinase-dependent regulation is critical to trigger PrimPol-dependent repriming. Accordingly, PrimPol is phosphorylated at Ser255 under KRAS^G12V-induced RS, consistent with a recent study showing that chemically induced RS induces PrimPol phosphorylation at the same site[33], indicating that ATR-Chk1 signaling-dependent phosphorylation of PrimPol at Ser255 is a critical switch to turn on its repriming activity. Since Ser255 in PrimPol is located in the unstructured/disordered loop region and is conserved only among higher primates, multiple layers of mechanisms, including PrimPol gene expression, protein stability, recruitment when needed, and/or alternative mechanisms, enable the control of PrimPol activity in other organisms[30,45,69,70].

KRAS^G12V expression slows replication forks in a manner dependent on RNA transcription but independent of the well-characterized events of R-loop formation and TRCs[12,48,71,72]. Expression of KRAS^G12V may promote global chromatin decompaction[73] and generate RNAs, including non-coding RNAs or transcripts from certain genes, that promote PRC2 recruitment and H3K27me3. This modulated chromatin leads to the formation of HP1α-and HP1β-mediated facultative heterochromatin, abrogating replication fork progression and causing fork reversal. Several proteins, including SMARCAL1, ZRANB3, and HTLF, catalyze the formation of reversed replication forks[74]. In particular, SMARCAL1 is required to cope with endogenous sources of RS in the absence of exogenous drugs[75], and ATR phosphorylates SMARCAL1 on Ser652 to negatively regulate its fork remodeling activity[43]. In the presence of increased ATR expression, locally compacted heterochromatin was skipped by PrimPol-dependent repriming, leaving ssDNA gaps behind forks. These findings suggest that ATR, via its kinase activity, not only regulates PrimPol activity but also prevents fork reversal to promote PrimPol-mediated repriming during KRAS^G12V-induced RS.

What is the consequence of ATR-PrimPol-mediated RST during KRAS^G12V-induced RS? First, more cells survive and acquire anchorage-independent growth. Second, these cells exhibit more genomic instability. These two phenotypes are clearly suppressed by ATRi treatment, which also suppresses PrimPol-mediated repriming, suggesting that ATR-PrimPol-mediated RST promotes the completion of replication with ssDNA gaps in the genome, leading to elevated genomic instability that might provide a driving force for acquisition of genomic alterations required for clonal expansion, eventual tumorigenesis and malignant transformation. Consistently, among patients with KRAS-mutant tumors, patients with high expression of ATR or both ATR and PrimPol had a significantly lower OS rate, indicating that those tumors may gain diverse resistance mechanisms to anticancer drug therapy, which may lead to poor prognosis. Notably, the poor prognosis of patients with high ATR and PrimPol expression was observed only for KRAS-mutant tumors and not for other tumors, suggesting a unique link between oncogenic KRAS and ATR via RS.

ATR inhibitors are a new class of anticancer compounds that are in early phase clinical trials[76–79]. Among the combination therapies tested, PARP inhibitors (PARPis) are some of the most promising synergistic partners of ATRi[80–82]. PARPis induce ssDNA gaps on the leading strand behind replication forks via PrimPol-mediated repriming[31,45,83,84] and on the lagging strand via defects in Okazaki fragment processing[85]. Defects in BRCA1/2 and Rad51 cause not only defects in homologous recombination (HR) but also failure to fill ssDNA gaps generated by PrimPol, conferring ATRi susceptibility[46]. These findings suggest that ssDNA gaps generated in the genome might be a prerequisite for sensitization to ATRi. Furthermore, the ATR pathway is required for the PrimPol-dependent adaptive response to cisplatin[45], and the observed notable benefit of combining an ATRi with gemcitabine in patients with a platinum-free interval of 3 months or less but not in patients with an interval >3 months[86] suggests that activation of the ATR-PrimPol pathway in response to RS is the key determinant of successful ATRi therapy. Therefore, our findings that ATR-PrimPol mediates RST in response to KRAS^G12V-induced RS provide a scientific basis for targeting KRAS mutations in LUAD via pharmacological inhibition of ATR activity.

## Methods
### Cell culture and cell lines
HEK293T cells were cultured in Dulbecco's modified Eagle's medium (DMEM; Nacalai Tesque, Cat# 08459-64). RPE-1 cells (a kind gift from Dr. Hochegger) and derived cell lines were cultured in DMEM-Ham's F-12 medium (Nacalai Tesque, Cat# 11581-15). H3122 (a kind gift from Dr. Kobayashi), H1975, H1819, H358, A427 and H2009 cells were cultured in RPMI 1640 (Nacalai Tesque, Cat# 30264-56). These media were supplemented with 10% fetal bovine serum (Gibco), 100 U/ml

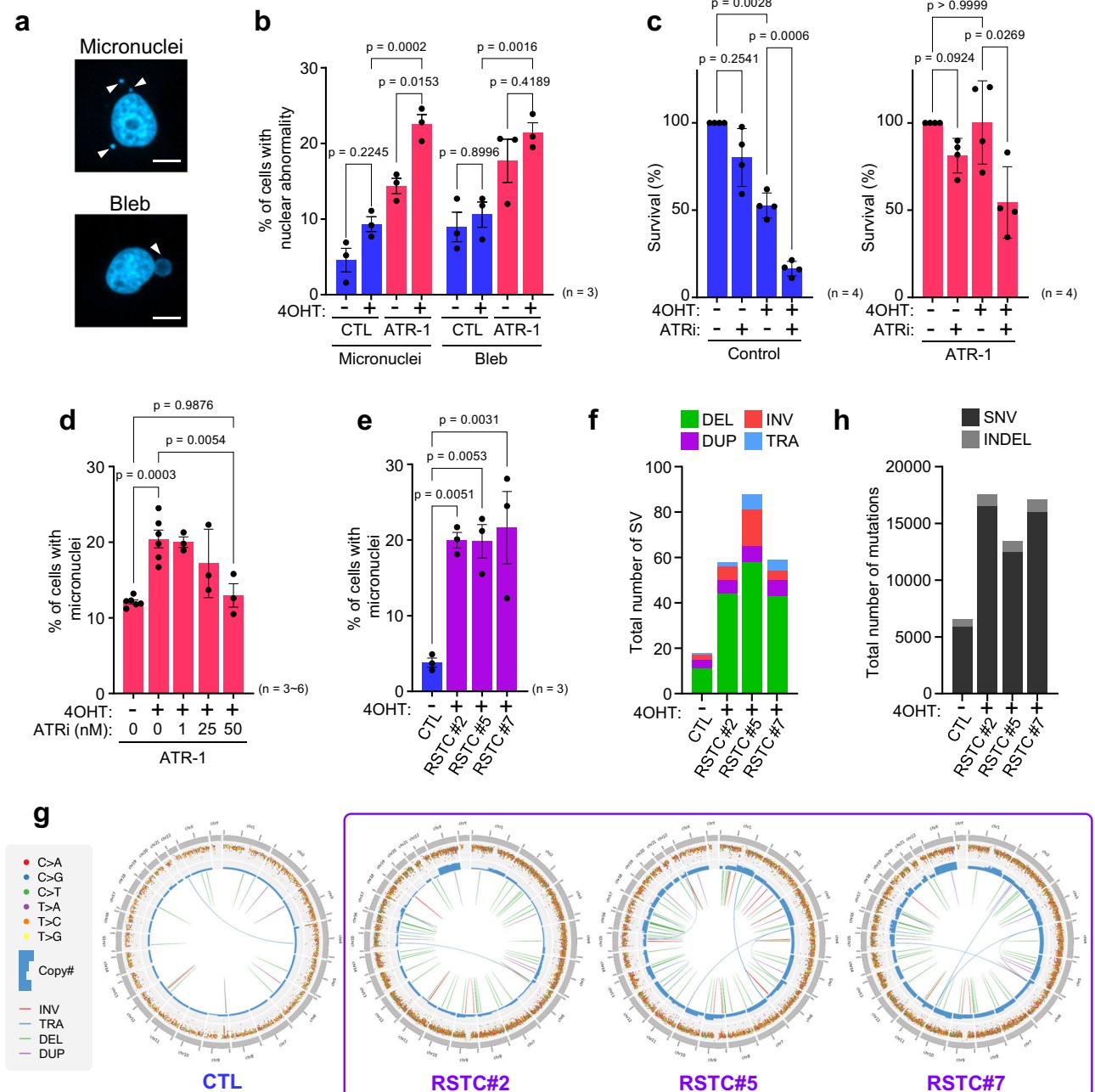

**Fig. 6 | Genomic instability driven by KRAS^G12V is tolerated in cells with elevated ATR expression. a** Representative image of micronuclei and bleb (white arrow). Scale bar = 10 μm. **b** Quantification of micronuclei and bleb positive cells in Control and ATR-1 cells treated with 0.1 μM of 4OHT for 3 days. The results represent the means ± SEM of three independent experiments. two-way ANOVA Tukey's test. **c** Quantification of the plate colony- forming efficiency assay with Control and ATR-1 cells treated with 0.1 μM of 4OHT and 25 nM of ATRi for 6 days. The results represent the means ± SEM of four independent experiments. one-way ANOVA Tukey's test. **d** Quantification of the micronuclei positive cells in ATR-1 cells treated with 0.1 μM of 4OHT and indicated low-concentration of ATRi for 3 days. The

results represent the means ± SEM of six independent experiments in the absence of ATRi, three independent experiments in the presence of ATRi. one-way ANOVA Tukey's test. **e** Quantification of micronuclei positive cells in RSTCs clone #2, #5, #7 cultured in 0.1 μM of 4OHT-containing medium. The results represent the means ± SEM of three independent experiments. one-way ANOVA Tukey's test. **f** Total number of structural variants (SV) in RSTCs clone #2, #5, #7. DEL, deletion; INV, inversion; DUP, duplication; TRA, translocation. **g** CIRCOS plot of RSTCs clone #2, #5, #7 revealed by whole genome sequence. **h** Total number of mutations in RSTCs clone #2, #5, #7. SNV, single nucleotide variants; INDEL, insertion and/or deletion. All source data are provided as a Source Data file.

penicillin and 100 μg/ml streptomycin (Nacalai Tesque, Cat# 26252-94). SAEC (human small airway epithelial cells, a kind gift from Dr. Kiyono) were immortalized via the expression of hTERT, a CDK4 mutant and cyclin D1. SAECs and derived cell lines were cultured in BronchiaLife™ Epithelial Basal Medium (Lifeline Cell Technology, Cat# LM-0007) supplemented with the components of a LifeFactors® Kit (Lifeline Cell Technology, Cat# LS-1047, containing human serum

albumin, lecithin, linoleic acid, L-glutamine, bovine pituitary extract, and TM-1 factor). All cells were cultured at 37 °C in 5% $CO_2$.

**Generation of the cell line with inducible KRAS^G12V expression**
To establish 4OHT-inducible KRAS^G12V cells, HEK293T cells were cotransfected with 0.5 μg/ml pQCXIHyg-mERT2-KRAS^G12V DNA construct (provided by Dr. Kiyono) and the viral packaging and viral

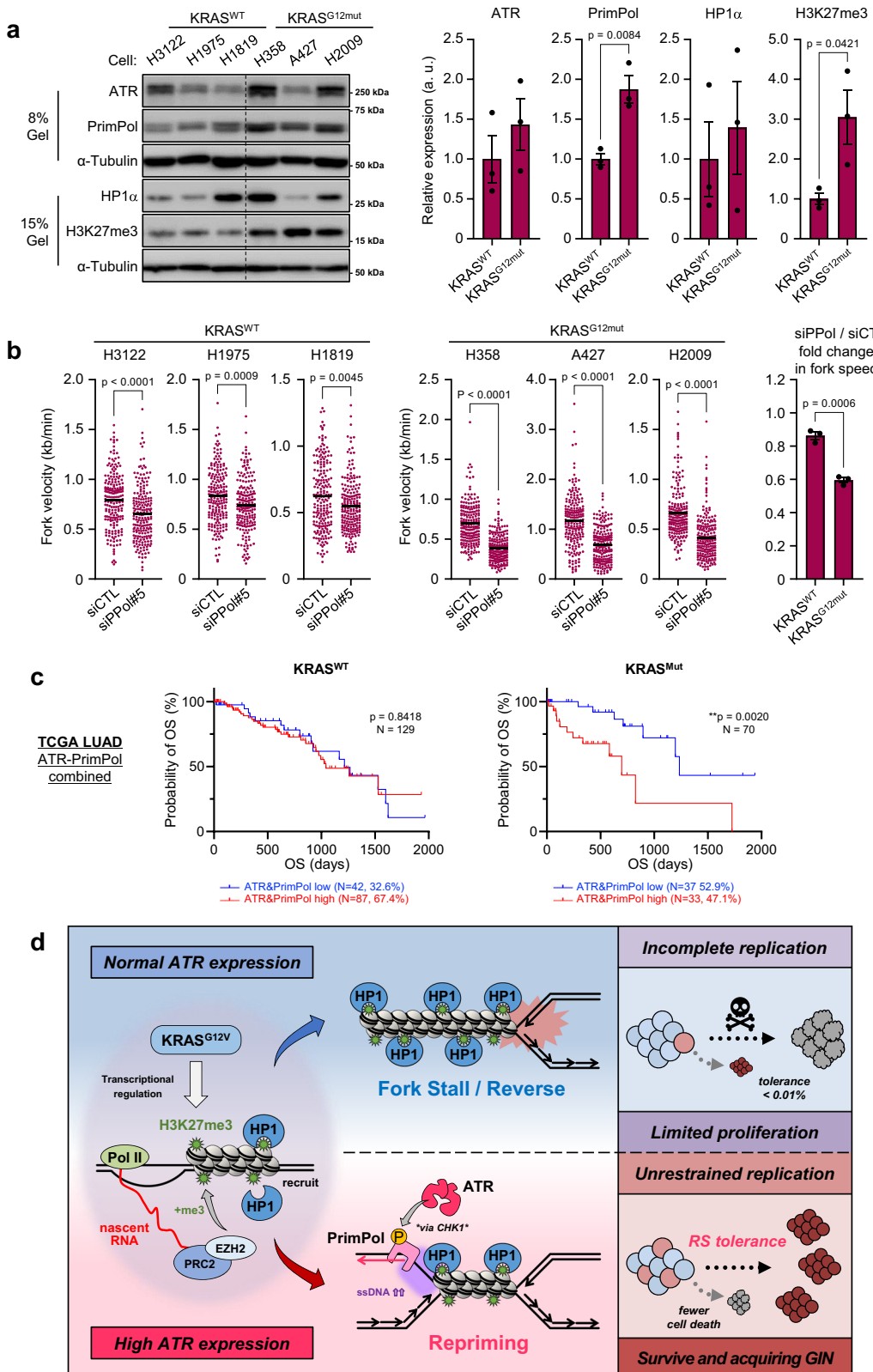

envelope plasmids pCL-GagPol and pEF6/env (10A1), respectively, using Polyethyleneimine (PEI) "MAX" (PEI; Polyscience, Cat# 24765). The retroviral supernatant was collected, filtered (0.45 mm PVDF) and concentrated using a Retro-X Concentrator (TaKaRa, Cat# 631456). Concentrated virus was resuspended in medium containing 8 µg/ml polybrene (Sigma–Aldrich, Cat# H9268) for infection of SAECs or RPE-1 cells.

### Generation of cell lines with constitutive ATR expression and inducible PrimPol, RNaseH1 or HP1α expression

To generate cells with constitutive ATR expression, ATR cDNA was amplified by PCR using Pfu Turbo DNA Polymerase (Agilent, Cat# 600250) and inserted into the NotI and NheI sites of CSII-CMV-MCS-IRES2-Bsd (Riken, Cat# RDB04385) using the In-Fusion HD cloning protocol (TaKaRa, Cat# 639649). HEK293T cells were

**Fig. 7 | Proposed model for ATR-PrimPol-mediated RST under KRAS^G12V-induced RS. a** Left, *KRAS^WT* cells (H3122, H1975, H1819) and *KRAS^G12mut* cells (H358, A427, H2009) were incubated for 2 days. The indicated proteins expression levels were analyzed by western blot analysis. Right, quantification of each expression level. The results represent the means ± SEM of three cells. Unpaired *t*-test. **b** Left, Dot plot and mean of fork speed in *KRAS^WT* cells and *KRAS^G12mut* cells transfected with 1 nM of siControl or siPPol#5 for 24 h as shown in (**S7b**). Representative result of two independent reproducible experiments are shown. Black lines indicate the mean; n = 200; two-tailed Mann−Whitney *t*-test. Right, PrimPol KD fold change in fork speed. The results represent the means ± SEM of three cells. two-tailed unpaired *t*-test. **c** High expression of both ATR and PrimPol is associated to poor prognosis of LUAD patients. Overall Survival (OS) according to ATR and PrimPol mRNA expression from totally 199 of LUAD patients harboring *KRAS^WT* and *KRAS^mut* were sanalyzed. Log-rank *p*-values are shown. **d** In the early step of KRAS^G12V expression, KRAS^G12V induces transcription-dependent locally compacted heterochromatin region mediated by H3K27me3, leading to RS. When cells express normal level of ATR, replication forks stall, causing incomplete replication. In contrast, when cells express high level of ATR, ATR-PrimPol pathway functions as a regulatory module at replication forks to complete replication by promoting repriming, allowing cells to survive under KRAS^G12V-induced RS with acquiring genomic instability. All source data are provided as a Source Data file.

cotransfected with 0.5 μg/ml CSII-CMV-ATR-IRES2-Bsd DNA construct, and the viral packaging and viral envelope plasmids pCAG-HIVgp and pCMV-VSV-G-RSV-Rev, respectively, using PEI. The viral supernatant was collected, filtered (0.45 mm PVDF) and concentrated using a Lenti-X Concentrator (TaKaRa, Cat# 631232). Concentrated virus was resuspended in medium containing 8 μg/ml polybrene for infection of SAECs or RPE-1 cells. CSII-ATR-transduced single-cell clones were selected before retroviral infection for KRAS^G12V expression.

To establish doxycycline-inducible PrimPol cells, PrimPol-myc cDNA (OriGene, Cat# rc209004) was amplified by PCR using Pfu Turbo DNA Polymerase and inserted into the BamHI and NotI sites of the pENTR GateWay vector using the In-Fusion HD cloning protocol. pENTR-PrimPol-myc was recombined into CSIV-TRE-RfA-UbC-KT-Puro (a kind gift from Dr. Nakanishi) using the GateWay system (Invitrogen, Cat# 11791-020). HEK293T cells were cotransfected with 0.5 μg/ml CSIV-TRE-PrimPol-myc-RfA-UbC-KT-Puro, pCAG-HIVgp and pCMV-VSV-G-RSV-Rev plasmids using PEI. The virus supernatant was collected, filtered and concentrated using a Lenti-X Concentrator. Concentrated virus was resuspended in medium containing 8 μg/ml polybrene to infect SAECs.

To establish doxycycline-inducible RNaseH1 cells, HEK293T cells were cotransfected with 0.5 μg/ml each of the pInd20-TRE-RNaseH1-GFP-UbC-IRES-Neo, pCAG-HIVgp and pCMV-VSV-G-RSV-Rev plasmids using PEI. The viral supernatant was collected, filtered and concentrated using a Lenti-X Concentrator. Concentrated virus was resuspended in medium containing 8 μg/ml polybrene for infection of SAECs.

To establish doxycycline-inducible HP1α cells, myc-HP1α cDNA (IDT, custom designed) was inserted into the BsmBI and XhoI sites of pInd20-TRE-MCS-UbC-IRES-Neo using the In-Fusion HD cloning protocol. HEK293T cells were cotransfected with 0.5 μg/ml each of the pInd20-TRE-myc-HP1α-UbC-IRES-Neo, pCAG-HIVgp and pCMV-VSV-G-RSV-Rev plasmids using PEI. The viral supernatant was collected, filtered and concentrated using a Lenti-X Concentrator. Concentrated virus was resuspended in medium containing 8 μg/ml polybrene for infection of SAECs.

### Exogenous gene expression and drug treatments
The following antibiotics were administered at the indicated concentrations: 50 μg/ml hygromycin B (Wako, Cat# 080-07683) for pQCXIHyg-mERT2-KRAS^G12V vector selection, 10 μg/ml blasticidin S (Wako, Cat# 022-18713) for CSII-CMV-ATR-IRES2-Bsd vector selection, 1 μg/ml puromycin (InvivoGen, Cat# ant-pr-1) for CSIV-TRE-PrimPol-myc-RfA-UbC-KT-Puro vector selection, and 200 μg/ml G418 (InvivoGen, Cat# ant-gn-1) for pInd20-TRE-RNaseH1-GFP-UbC-IRES-Neo vector and pInd20-TRE-myc-HP1α-UbC-IRES-Neo vector selection. Antibiotics were added to the media.

4OHT (Abcam, Cat# ab141943) was dissolved in DMSO to a 0.1 mM stock solution and stored at −20 °C. For 4OHT-inducible KRAS^G12V expression, cells were treated with 0.1 μM 4OHT for the indicated time periods. Doxycycline (Sigma−Aldrich, Cat# D9891-1G) was dissolved in dH₂O to a concentration of 1 mg/ml. For doxycycline-inducible PrimPol, RNaseH1, and HP1α expression, cells were treated with 1 μg/ml doxycycline for the indicated time periods.

The ATR inhibitor berzosertib (Selleck, Cat# S7102, 1 mM stock) and the Chk1 inhibitor rabusertib (Selleck, Cat# S2626, 1 mM stock) were dissolved in DMSO and administered at the indicated final concentrations (see figure). Cycloheximide (CHX; Wako, Cat# 037-20991, 10 mg/ml stock) was dissolved in dH₂O and administered at a final concentration of 300 μg/ml. Roscovitine (Calbiochem, Cat# 557360-5MG, 10 mM stock) was dissolved in DMSO and administered at a final concentration of 25 μM. 5,6-Dichloro-1-b-D-ribofuranosyl-1H-benzimidazole (DRB; Sigma−Aldrich, Cat# D1916-10MG, 100 mM stock) was dissolved in DMSO and administered at a final concentration of 100 μM. Chloroquine (CQ; Sigma−Aldrich, Cat# C6628-25G) was dissolved in dH₂O to a concentration of 4 mM (freshly prepared at the time of each experiment) and administered at a final concentration of 40 μM. GSK126 (Selleck, Cat# S7061, 5 mM stock) was dissolved in DMSO and administered at a final concentration of 2.5 μM.

PhosSTOP (Sigma−Aldrich, Cat# 04906837001) and cOmplete (Sigma−Aldrich, Cat# 11697498001) were dissolved in the in vitro buffers according to the manufacturer's instructions.

### Gene silencing with RNAi
Cells were transfected with 1 nM (Ambion products) or 5 nM (Integrated DNA Technologies (IDT) products) siRNAs by the reverse transfection method with Lipofectamine RNAiMAX (Invitrogen, Cat# 13778-150) according to the manufacturer's instructions. After 18 - 24 h of incubation with siRNAs, the cell medium was replaced with fresh medium. The following siRNAs were purchased from Thermo Fisher Scientific: siPRIMPOL (Ambion, ID# s47417: siPP#3 and ID# s47418: siPP#2, ID# s47416:siPP#1); siHP1α (Ambion, ID# s23883: siHP1α#1 and ID# s23884: siHP1α#2); siHP1β (Ambion, ID# s21549: siHP1β#1 and ID# s21550: siHP1β#2). siUTR-PRIMPOL (Ambion, sense: GUCUGUGA-GAUUUGAUAAAAtt, antisense: UUUAUCAAAUCUCACAGACaU; siPP#5) and siUTR-HP1α (Ambion, sense: GUUGCCCAUCUGUUAAAAAtt, antisense: UUUUUAACAGAUGGGCAACaU; siHP1α#3) were custom designed. Silencer Select Negative Control siRNA (Ambion, Cat# 4390843) was used as a control siRNA. siUTR-PRIMPOL (IDT, ID# hs.Ri.PRIMPOL.13.3: siPPol#4) and IDT negative control siRNA (IDT, Cat# 51-01-14-03) were purchased from IDT.

### Correlation analysis of RNA expression and patient survival
The RNA-seq and clinical data related to LUAD samples provided by the TCGA project were obtained from the Genomic Data Commons Data Portal. RNA-seq and clinical data related to LUAD samples were also obtained from the National Cancer Center Biobank (NCC, Japan).

RNA-seq of samples from the NCC Biobank was conducted using 1.1 μg of RNA isolated from snap-frozen cancer tissue samples obtained from 916 patients. After quality assessment (RIN > 6.0) with an Agilent 2100 Bioanalyzer (Agilent Technologies, Santa Clara, CA, USA), polyadenylated RNA libraries were generated using the TruSeq Stranded mRNA Library Prep Kit (Illumina) and sequenced on the Illumina HiSeq 2500 platform using 2 × 75 bp/2 × 100 bp paired-end reads or on the NovaSeq 6000 platform. Read mapping was performed using STAR

version 2.4.2a[87] with the human genome (GRCh38) (https://gdc.cancer.gov/about-data/gdc-data-processing/gdc-reference-files) and transcriptome data (GENCODE version 31[88]) as reference datasets. Transcripts per million (TPM) values were calculated using the StringTie (2.0.4) program[89]. This study was approved by the Institutional Review Board of the National Cancer Center.

The samples were grouped depending on ATR expression, PrimPol expression, KRAS mutation status, and the previous/current smoking status of the patient. The low and high expression groups for each gene were defined as follows. The cohort was divided into two groups (High and Low) such that the average expression levels in the Low group (Av_Low) and High group (Av_High) met the following condition: the ratio of the average expression levels [(Av_High)/(Av_Low)] is minimized (typically ~2) or not <1.5. The OS of each group was analyzed by the Kaplan–Meier method, with the log-rank (Mantel–Cox) test.

### Western blot analysis

Cells were lysed in sampling buffer (120 mM Tris-HCl (pH 6.8), 12% glycerol, 4% SDS, 200 mM DTT, 0.004% bromophenol blue) and denatured at 95 °C for 5 min. The protein concentration in the cell lysate was estimated by the Bradford assay (XL-Bradford, Apro Science, Cat# KY-1031). Equal amounts of protein were separated by SDS–PAGE (10% SDS-polyacrylamide gel) at 120 V for 2 h and were then transferred to polyvinylidene difluoride (PVDF) membranes (Immobilon-P, Millipore, Cat# IPVH00010) at 150 mA for ~16 h at 4 °C. The membranes were blocked with 5% skim milk in TBST (25 mM Tris-HCl (pH 7.8), 140 mM NaCl, 0.1% Tween 20) for 20 min at room temperature (RT) and probed with the following primary antibodies diluted in 5% skim milk in TBST (supplemented with 0.02% of NaN$_3$): anti-pan-RAS (1:1000, Santa Cruz, Cat# sc-166691), anti-αTubulin (1:5000, MBL, Cat# PM054), anti-phospho-ATR (Thr1989) (1:3000, Cell Signaling Technology, Cat# 30632), anti-ATR (1:3000, GeneTex, Cat# GTX128146), anti-phospho-Chk1 (Ser317) (1:500, Cell Signaling Technology, Cat# 12302), anti-phospho-Chk1 (Ser345) (1:500, Cell Signaling Technology, Cat# 2348), anti-Chk1 (1:100, Santa Cruz, Cat# sc-8408), anti-phospho-RB (Ser807/811) (1:1000, Cell Signaling Technology, Cat# 8516), anti-E-cadherin (1:1000, Cell Signaling Technology, Cat# 3195), anti-Vimentin (1:1000, Cell Signaling Technology, Cat# 5741), anti-PrimPol (1:3000, Proteintech, Cat# 29824-1-AP), anti-myc tag (1:1000, MBL, Cat# 192-3), anti-HistoneH1.4 (1:1000, Cell Signaling Technology, Cat# 41328), anti-GAPDH (1:1000, Wako, Cat# 010-25521), anti-phospho-Chk1 (Ser296) (1:500, Cell Signaling Technology, Cat# 90178), anti-RNaseH1 (1:1000, Abcam, Cat# ab56560), anti-HP1α (1:1000, Millipore, Cat# 05-689), anti-HP1β (1:1000, Cell Signaling Technology, Cat# 8676). After 3 washes with TBST, the membranes were incubated with Horseradish Peroxidase (HRP)-conjugated AffiniPure Goat Anti-Rabbit or Mouse IgG (H + L) (1:5000, Jackson ImmunoResearch, Cat# 111-035-003 or Cat# 115-035-003) in 5% skim milk for 1 h at RT and were then washed with TBST 5 times. The blots were developed with Western Lightning Plus ECL Reagent (PerkinElmer, Cat# NEL105001EA) according to the manufacturer's instructions and imaged with an LAS 3000 luminescent image analyzer (Fujifilm).

### Long-term time-to-progression (TTP) assay

A total of $2 \times 10^5$ cells were plated in 6-cm plates with or without 0.1 μM 4OHT. After 3 or 4 days of incubation, cells were collected and counted. This 2D culture process was repeated for ~30 days depending on the deleterious effects of 4OHT-induced KRAS$^{G12V}$. The daily growth rate (μ) was calculated according to the formula: $\mu = (ln\ N - ln\ N')/t$, where $N$ is the number of cells at the current week, $N'$ is the number of cells plated at the previous week, and $t$ is the culture days. Based on the daily growth rate (μ) calculated at every split time, the total current cell number ($Nx$) was estimated according to the formula $ln\ Nx = ln\ N_O + \mu *$

$T$, where $N_O$ is the number of cells plated at the initial split point and $T$ is the total number of incubation days.

### Anchorage-independent growth assay

A total of $2 \times 10^4$ cells were suspended in 1 ml of 0.4% soft agar solution (SeaPlaque agarose, Lonza, SeaPlaque™, Cat# 50101) and then plated on 1.5 ml of 0.7% bottom agar. Top medium containing 0.1 μM of 4OHT was added every 3 ~ 4 days. After ~30 days of culture at 37 °C, cells were stained with 0.01% crystal violet dissolved in 20% MtOH for 1 h at RT and were then imaged using a LAS-3000 imager (Fujifilm).

### RNA extraction and real-time PCR

A total of $1 \times 10^5$ cells were plated in 6-well plates. Following treatments, total RNA was purified using ISOGEN II (Nippon Gene, Cat# 311-07361) according to the manufacturer's instructions. Extracted RNA was resuspended in TE buffer (pH 8.0, NipponGene, Cat# 314-90021).

To detect ATR, PrimPol and ActinB mRNA, reverse transcription of the mRNA library was carried out using a High-Capacity RNA-to-cDNA Kit (Applied Biosystems, Cat# 4387406) according to the manufacturer's instructions. SYBR Green real-time PCR was performed using a KAPA SYBR FAST ABI Prism qPCR Kit (Kapa Biosystems, Cat# KK4604). The following primers were used at 500 nM: ATR-F, 5'-GGTATGCTCTCACTTCCATG-3'; ATR-R, 5'-GTCAGAAGAAACACA-CATCG-3'; PRIMPOL-F, 5'-TGTGGCTTTGGAGGTTACTGA-3'; PRIMPOL-R, 5'-TTCTACTGAAGTGCCGATACTGT-3'; Hs_ACTB_1_SG QuantiTect Primer Assay (QIAGEN, ID# QT00095431).

To detect miR185, reverse transcription was carried out using a TaqMan™ MicroRNA Reverse Transcription Kit (Applied Biosystems, Cat# 4366596) and TaqMan™ MicroRNA Assays (Applied Biosystems, RNU48: Assay ID# 001006, miR185: Assay ID# 002271) according to the manufacturer's instructions. TaqMan real-time PCR was performed using TaqMan™ Fast Advanced Master Mix (Applied Biosystems, Cat# 4444556) according to the manufacturer's instructions.

PCR was performed using a StepOnePlus thermocycler (Applied Biosystem). The comparative Ct (ΔΔCt) method was selected to validate the quantification of mRNA or miRNA expression.

### Proliferation assay

A total of 2000 cells were seeded in 96-well plates and treated with 0.1 μM 4OHT. The numbers of proliferating cell were determined 7 days after 4OHT treatment using PrestoBlue™. According to the manufacturer's instructions, PrestoBlue™ Cell Viability Reagent (Invitrogen, Cat# A13262) was added directly to the cells and incubated for 1 h at 37 °C. The optical density (excitation: 560 nm, emission: 590 nm) was measured with a Biotek Epoch microplate reader and analyzed with Gen5 2.05 software (Biotek).

For HP1α KD effect validation, $1.5 \times 10^5$ cells were transfected with 1 nM siRNA for 2 days and then replated on 96-well plates with 0.1 μM 4OHT. After 5 days of incubation, cell proliferation was evaluated by Cell Titer-Glo 2.0 Assay (Promega, Cat# G9243) according to the manufacturer's instructions. The luminescence was measured with a Biotek Epoch microplate reader and analyzed with Gen5 2.05 software (Biotek).

### DNA fiber assay and S1 fiber assay

The standard DNA fiber assay was previously described[53]. A total of $1 \times 10^5$ cells were plated in 6-well plates. Following treatments, cells were sequentially labeled with 100 μM IdU (5-iodo-2'-deoxyuridine, Sigma–Aldrich, Cat# l7125-5G) for 20 min, washed with PBS 3 times, labeled with 100 μM CldU (5-chloro-2'-deoxyuridine, Sigma–Aldrich, Cat# C6891-100MG) for 40 min, and washed with PBS. The labeled cells were trypsinized and pelleted at 800 x g for 5 min at 4 °C and were then resuspended in 200 μl of PBS.

The S1 fiber assay was performed as previously described with modifications[45]. Labeled cells were permeabilized with CSK100 (10 mM

MOPS (pH 7.0), 100 mM NaCl, 3 mM MgCl$_2$, 300 mM sucrose, 0.5% Triton X-100) for 10 min at RT, gently washed twice with S1 buffer (30 mM sodium acetate (pH 4.6), 50 mM NaCl, 10 mM zinc acetate, 5% glycerol), treated with 10 U/ml S1 nuclease (Invitrogen, Cat# 18001-016) in S1 buffer for 30 min at 37 °C, and collected in 500 μl of PBS containing 1% BSA (Nacalai Tesque, Cat# 01860-07) with a cell scraper. Cell nuclei were pelleted at 4900 x *g* for 5 min at 4 °C, and 300 μl of the supernatant was then removed, leaving 200 μl. Cell nuclei were carefully resuspended by pipetting.

For both the standard DNA fiber assay and S1 fiber assay, 100 μl of fixative solution (3:1 MtOH:acetic acid) was added dropwise to 200 μl of cell suspension. This step was repeated 3 times, and 500 μl of fixative solution was then added; This step was repeated 2 times. Cells/nuclei were pelleted at 800 x *g* for 5 min at 4 °C and resuspended in fixative solution to a final concentration of ~1000 cells/μl.

Fifty microliters of fixed cells were spotted onto slides (Matsunami, Cat# S2112). After drying for 3 min, the slides were immersed in lysis solution (200 mM Tris-HCl (pH 7.5), 50 mM EDTA, 0.5% SDS) for ~20 min at 37 °C. The DNA fibers released from the cells were extended by tilting the slides in a high-humidity chamber for at least 30 min. The slides were immersed in fixative solution for 20 min and were then dried overnight (O/N) at 4 °C.

For immunostaining of DNA fibers, DNA fibers were rehydrated in PBS twice for 5 min each, immersed in 2.5 M HCl for 1 h to denature the DNA and washed with PBS 3 times for 5 min each. After blocking (5% BSA and 0.1% Tween 20 in PBS) for 1 h, the slides were incubated for ~2 h at 37 °C with anti-IdU (1:25, BD Biosciences, 347580) and anti-CldU (1:100, Abcam, ab6326) antibodies diluted in blocking solution to label the DNA, washed 3 times with PBST (PBS containing 0.1% Tween 20) and incubated for 1 h with Alexa Fluor 488-conjugated anti-rat IgG (1:500, Invitrogen, Cat# A11006) and Alexa Fluor 555-conjugated anti-mouse IgG (1:250, Invitrogen, Cat# A21422) in blocking solution. The stained slides were washed with PBST 3 times and mounted with ProLong Gold Antifade (Invitrogen, Cat# P36980).

Images were acquired with a fluorescence microscope (Celldiscoverer 7, Zeiss) and analyzed using ImageJ software (v2.0.0-rc-69, National Institutes of Health). Continuous red (IdU) and green (CldU) tracks were measured, and the distance in micrometers was converted to kilobase pairs (1 μm = 3.5 kb). Finally, the fork speed was calculated from the labeling time. A total of 200 - 500 fibers were measured in each sample, and 200 randomly selected fibers are shown in the figures.

## Immunofluorescence staining

A total of $1 \times 10^4$ cells were plated in collagen-coated 8-well chamber dishes. Following treatments, cells were prefixed with ice-cold PBS for 5 min on ice, and a pre-extraction step was performed with pre-extraction buffer (20 mM MOPS (pH 7.0) and 0.05% Triton X-100 in PBS) for 5 min on ice. Cells were fixed with 4% paraformaldehyde (PFA) in PBS for 10 min prior to post-extraction with 0.25% Triton X-100 in PBS for 10 min. To detect incorporated native BrdU, cells were further fixed by ice-cold MtOH for 10 min. After blocking with wash buffer (3% BSA and 0.05% Tween 20 in PBS) for 1 h at 37 °C, the cells were incubated for 1 h at 37 °C or O/N at 4 °C with the following primary antibodies in wash buffer: anti-BrdU (1:1000, GE health, Cat# RPN20AB) anti-RPA32 (1:200, Abcam, Cat# ab2175), anti-phospho-RPA32 (Ser4/8) (1:200, Bethyl, # A300-245A), anti-phospho-RPA32 (Ser33) (1:200, Bethyl, # A300-246A), anti-γH2AX (1:500, Cell Signaling Technology, Cat# 9718), anti-H3K27me3 (1:500, Cell Signaling Technology, Cat# 9733), H3K9me3 (1:500, Abcam, Cat# ab8898), anti-PICH (1:250, Abnova, Cat# H00054821-B01P), and anti-phospho-HistoneH3 (Ser10) (1:500, Millipore, Cat# 06-570). The cells were washed with wash buffer 3 times and incubated with the following Alexa Fluor-conjugated secondary antibodies for 1 h at RT: Alexa 488 donkey anti-rabbit or anti-mouse IgG (H + L) (1:500, Jackson ImmunoResearch,

Cat# 711-545-152 or 715-545-151) and Alexa 594 donkey anti-rabbit or anti-mouse IgG (H + L) (1:500, Jackson ImmunoResearch, Cat# 711-585-152 or 715-585-151). After 3 washes, the nuclei were stained with 1 μg/ml DAPI and mounted with VECTASHIELD. Immunofluorescence images were acquired with a fluorescence microscope (Celldiscoverer 7, Zeiss) for intensity analysis (Zen v2.6, Zeiss).

## PrimPol phosphoproteomics assay

Following treatments, $1.5 \times 10^7$ cells were washed with PBS supplemented with PhosSTOP and were then lysed in 8 ml of lysis buffer (30 mM Tris-HCl (pH 7.5), 150 mM NaCl, 2.5 mM MgCl$_2$, 0.5% NP40, PhosSTOP, cOmplete, 1 mM dithiothreitol (DTT)). After treatment with 100 U/ml benzonase for 1 h at 4 °C, the insoluble fraction was pelleted at $20000 \times g$ for 10 min at 4 °C. The cell lysate was collected, and the protein concentration was estimated by the Bradford assay. Samples containing equal amounts of protein were incubated with an anti-myc tag antibody (15 μg/$1.5 \times 10^7$ cells, MBL, Cat# M192-3) conjugated to Protein-G Sepharose beads (Cytiva, Cat# 17061802) for 4 h at 4 °C, and the beads were washed 3 times with lysis buffer. Immunoprecipitated proteins were denatured with sampling buffer and were then separated by SDS–PAGE. Proteins were stained using SYPRO Ruby Protein Gel Stain (Invitrogen, Cat# S12000) according to the manufacturer's instructions.

Protein bands were excised and subjected to in-gel tryptic digestion essentially as described previously[90]. In brief, the gel pieces were destained and washed, and after DTT reduction and iodoacetamide alkylation, the proteins were digested with trypsin O/N at 37 °C. The resulting tryptic peptides were extracted from the gel pieces by sequential treatment with 30% acetonitrile, 0.3% trifluoroacetic acid and 100% acetonitrile. The extracts were evaporated in a vacuum centrifuge to remove the organic solvent and were then desalted and concentrated with reversed-phase C18 StageTips as previously described[91]. Immobilized metal affinity chromatography (IMAC) enrichment of phosphopeptides was performed as previously described[92].

LC–MS/MS was performed by coupling an UltiMate 3000 Nano LC system (Thermo Fisher Scientific) and an HTC-PAL autosampler (CTC Analytics) to a Q Exactive hybrid quadrupole-Orbitrap mass spectrometer (Thermo Fisher Scientific). Peptides were delivered to an analytical column (75 μm × 30 cm, packed in-house with ReproSil-Pur C18-AQ, 1.9 μm resin, Dr. Maisch, Ammerbuch, Germany) and separated at a flow rate of 280 nl/min using an 85-min gradient from 5% to 30% of solvent B (solvent A, 0.1% formic acid (FA) and 2% acetonitrile; solvent B, 0.1% FA and 90% acetonitrile). The Q Exactive instrument was operated in data-dependent mode. Survey full-scan MS spectra (m/z 350 to 1800) were acquired in the Orbitrap at a resolution of 70,000 after ion accumulation to a target value of $3 \times 10^6$. The dynamic exclusion time was set to 20 s. The 12 most intense multiply charged ions ($z \geq 2$) were sequentially accumulated to a target value of $1 \times 10^5$ and fragmented in the collision cell by higher-energy collisional dissociation (HCD) with a maximum injection time of 120 ms and a resolution of 35,000. Typical mass spectrometric conditions were as follows: spray voltage, 2 kV; heated capillary temperature, 250 °C; normalized HCD collision energy, 25%. The MS/MS ion selection threshold was set to $2.5 \times 10^4$ counts. A 2.0 Da isolation width was chosen.

Raw MS data were processed with MaxQuant (version 1.6.14.0) supported by the Andromeda search engine. The MS/MS spectra were searched against the UniProt human database (https://www.uniprot.org) with the following search parameters: full tryptic specificity, up to two missed cleavage sites, carbamidomethylation of cysteine residues as a fixed modification, and serine, threonine, and tyrosine phosphorylation, N-terminal protein acetylation and methionine oxidation as variable modifications. The false discovery rates (FDRs) of protein groups, peptides, and phosphosites were <0.01. The quantitative

values of the phosphorylation sites across the fractions were automatically integrated and summarized in "Phospho (STY) Sites.txt" by MaxQuant. Peptides identified from the reversed database or identified as potential contaminants were not used subsequent analysis.

## Chromatin fractionation
The fractionation method was previously described[93]. A total of $2 \times 10^5$ cells were plated in a 60 mm dish. Following treatments, cells were harvested with a scraper, pelleted at $800 \times g$ for 5 min at 4 °C, and resuspended in 120 µl of PBS. Approximately 15% of the cells were collected as the input sample. The cells were pelleted again and were then resuspended in 50 µl of Solution A (10 mM HEPES·KOH (pH 7.9), 10 mM KCl, 1.5 mM MgCl$_2$, 0.34 M sucrose, 10% glycerol, PhosSTOP, cOmplete, 1 mM DTT) supplemented with 0.2% Triton X-100. After incubation on ice for 5 min, the nuclear fraction was precipitated at 2500 x g for 5 min at 4 °C, and the supernatant was then collected as the cytosolic fraction. After washing with Solution A, the nuclear fraction was resuspended in 100 µl of Solution B (3 mM EDTA, 0.2 mM EGTA, 1 mM DTT) and incubated on ice for 20 min. The chromatin fraction was precipitated and washed with Solution B and finally pelleted at $20,000 \times g$ for 5 min at 4 °C. The fraction samples were denatured with sampling buffer, and the protein concentration was estimated by the Bradford assay. Samples containing equal amounts of protein were resolved by SDS–PAGE and analyzed by Western blotting.

## EU incorporation assay
EU incorporation assays were performed with a Click-iT RNA Alexa Fluor 594 Imaging Kit (Invitrogen, Cat# C10330). Following treatments, cells were labeled with 1 mM EU for 1 h or 23 h and were then fixed with 4% PFA in PBS for 15 min. Fixed cells were permeabilized with 0.5% Triton X-100 for 15 min. The Click-iT reaction was performed according to the manufacturer's instructions. Nuclei were stained with 1 µg/ml DAPI, and slides were mounted with VECTASHIELD (Vector Laboratories, Cat# H-1000). Images were acquired with a fluorescence microscope (Celldiscoverer 7, Zeiss) for intensity analysis (Zen v2.6, Zeiss).

## Slot blot assay
The DNA–RNA hybrid detection method was previously described (Matos et al., 2020). A total of $1 \times 10^5$ cells were plated in 6-well plates. Following treatments, cells were trypsinized, pelleted at $800 \times g$ for 5 min at 4 °C, and resuspended in 200 µl of PBS. Total DNA and RNA were purified using a QIAamp DNA Mini Kit (QIAGEN, Cat# 51106) according to the manufacturer's instructions and were then resuspended in 10 mM TE buffer (pH 8.0). To digest DNA–RNA hybrids, the nucleic acid solution was treated with 50 mU/µl RNaseH1 (NEB, Cat# M0297) for 1 h at 37 °C, and the reaction was then terminated with 5 mM EDTA. The DNA concentration was estimated using a NanoDrop 1000 spectrophotometer (Thermo Scientific). A total of 125 ng of nucleic acids was spotted on nylon membranes (Amersham Hybond-N, GE Healthcare, Cat# RPN303N) using a BIO-DOT SF apparatus (Bio-Rad) according to the manufacturer's instructions, and the membrane was subjected to UV crosslinking (120 mJ/cm$^2$). As mentioned in the Western blot method, the membrane was blocked and probed with S9.6 (1:500, Millipore, Cat# MABE1095) and anti-dsDNA (1:5000, Abcam, Cat# ab27156) primary antibodies diluted in Block Ace (DS Pharma Biomedical) prior to incubation with HRP-conjugated secondary antibodies and blot development with Western Lightning Plus ECL reagent. The S9.6 signals were normalized based on the dsDNA signals to determine the relative amount of DNA:RNA hybrids.

## MNase sensitivity assay
Cells ($8 \times 10^4$) were plated on a collagen-coated 35 mm dish. Following treatments, cells were permeabilized with CSK100 for 10 min at RT, gently washed with MNase buffer (50 mM Tris-HCl pH 8.0, 5 mM NaCl, 5 mM CaCl$_2$), and then treated with the 75 gelU/ml of MNase (New England Biolab, Cat# M0247S) in MNase buffer for 5 ~ 20 min at 25 °C. The MNase-treated cells were lysed in 250 µl of TES buffer (10 mM Tris-HCl pH 7.5, 10 mM EDTA, 0.5% SDS). The cell lysates were sequentially treated with 1 mg/ml RNaseA (NipponGene, Cat# 318-06391) for 45 min and with 200 µg/ml Proteinase K (QIAGEN, Cat# 19133) for 90 min at 50 °C, after which an equal volume of phenol/chloroform (Nippon-Gene, Cat# 311-90151) was added. The insoluble fraction was pelleted at 20,000 x g, the supernatant was mixed with 0.5 M of NaCl, and an EtOH wash was performed. The extracted DNA was resuspended in TE buffer (pH 8.0), resolved by 1.2% agarose gel (Fast Gene, Cat# NE-AG02) electrophoresis at 1 µg per lane, and stained with EtBr (NipponGene, Cat# 315-90051). Images were captured with a LAS 3000 luminescent image analyzer (FujiFilm) for intensity analysis with ImageJ.

## PLA
A total of $1 \times 10^4$ cells were plated in collagen-coated 8-well chamber dishes. To monitor the proximity between ssDNA gaps and H3K27me3, the cells were cultured with 10 µM BrdU (Sigma–Aldrich, Cat# B5002-1G) for 48 h and then incubated without BrdU for 24 h. S-phase cells were pulse labeled with 10 µM EdU for 30 min. As mentioned in the immunofluorescence method, cells were pre-extracted, fixed and processed with the Click-iT Plus EdU Alexa Fluor 488 Imaging Kit (Invitrogen, Cat# C10337) according to the manufacturer's instructions prior to blocking and primary antibody incubation O/N at 4 °C. PLA was performed using Duolink In Situ PLA kits according to the manufacturer's instructions (Sigma–Aldrich, Cat# DUO92001, Cat# DUO92005 and Cat# DUO92008) with the following primary antibodies: anti-BrdU (1:1000, GE Healthcare, Cat# RPN202) and anti-H3K27me3 (1:1600, Cell Signaling Technology, Cat# 9733) or anti-PCNA (1:500, Santa Cruz, Cat# sc-56) and anti-phospho-POLII (Ser2) (1:10000, Novus, Cat# NB100-1805). Nuclei were stained with 1 µg/ml DAPI, and slides were mounted with VECTASHIELD. Images were acquired with a fluorescence microscope (Celldiscoverer 7, Zeiss) for PLA focus counting (Zen v2.6, Zeiss) or a confocal laser scanning microscope for representative nucleus imaging (TCS SP8, Leica).

## Whole-transcriptome RNA-seq
Following treatments, $1 \times 10^6$ cells were washed with PBS, and total RNA was then purified using an RNeasy Plus Mini Kit (QIAGEN, Cat# 74134) according to the manufacturer's instructions. Paired-end sequencing with a read length of 150 bases was performed on the DNBSEQ-G400 (MGI tech) platform following the manufacturer's instructions. Raw sequence data quality was checked with FastQC software (ver. 0.11.9). Alignment and TPM calculations were performed using Kallisto with a human cDNA reference. Output files of Kallisto (ver. 0.46.0) were converted to an expression matrix with the 'tximport (ver. 1.18.0)' package in R software (ver. 3.6.3). Scaled TPM counts were used for further analysis. The distributions of H3K27me3 ChIP-seq signals within gene loci were visualized with the scale-regions function of deepTools[94]. H3K27me3 ChIP-seq data of SAECs were provided by Suzuki, A., et al. (2014)[49].

## Micronucleus assay
Following treatments, cells were fixed with 4% PFA in PBS and stained with an anti-phospho-HistoneH3 (Ser10) antibody (1:500, Millipore, Cat# 06-570) as described above. After imaging with a fluorescence microscope (Celldiscoverer 7, Zeiss), the number of cells with MN or blebs was determined. Phospho-HistoneH3 (Ser10)-positive cells were omitted because of nuclear membrane disappearance. At least 200 cells were analyzed per sample (Zen v3.1, Zeiss).

## Plate colony-forming efficiency assay
A total of $1 \times 10^4$ cells were plated in 6-well plates and treated with 4OHT and/or berzosertib for 6 days. The medium was changed 3 days

after cell seeding. Cells were then stained with 0.5% crystal violet dissolved in 20% MtOH for 30 min and rinsed 3 times with distilled water. After drying, images of each whole 6-well plate were acquired using a LAS-3000 imager (Fujifilm). The mean gray value of each well was quantified using ImageJ.

## WGS

Following treatments, ~$2 \times 10^5$ cells were lysed in 250 µl of TES buffer (10 mM Tris-HCl (pH 7.5), 10 mM EDTA, 0.5% SDS). Cell lysates were sequentially treated with 1 mg/ml RNaseA (NipponGene, Cat# 318-06391) for 45 min and 200 µg/ml proteinase K (QIAGEN, Cat# 19133) for 90 min at 50 °C prior to the addition of 260 µl phenol/chloroform (NipponGene, Cat# 311-90151). The insoluble fraction was pelleted at 20,000 x $g$, and the supernatant was mixed with 0.5 M NaCl and washed with EtOH 2 times. Extracted DNA was resuspended in 50 µl of TE buffer (pH 8.0).

Genome library preparation was performed with a TruSeq DNA PCR-Free Library Prep Kit (Illumina). Sequencing with 150 bp paired-end reads was performed on the NovaSeq 6000 system (Illumina) at a sequencing depth of ~120x, and the sequence reads were aligned to the human reference genome (hg38) with BWA-MEM (version 0.7.17)[95] according to the GATK Best Practices workflow. SAEC cells were defined as the normal-equivalent specimen, and Control-ER-Kras-$^{\text{G12V}}$, RSTC#2, RSTC#5 and RSTC#7 cells were defined as tumor-equivalent specimens. Somatic SNVs and small insertions and deletions (INDELs) were called with Mutect2 (gatk version 4.1.2.0)[96] with tumor-normal mode, subtracting variants in normal specimens from those in tumor specimens. These variants were filtered with FilterMutectCalls and FilterAlignmentArtifacts, and those with a variant allele frequency (VAF) of <0.1 were removed.

Structural variations were detected by the Manta program (version 1.6.0)[97], which identified the break-end junction by discordant paired and split reads. To identify high-confidence rearrangements, we selected them with the following criteria: PASS in the FILTER field, VAF > 0.1 and split read >0. Circos plots were generated with the shinyCircos program (ver 1)[98]. Allele-specific copy numbers, tumor purity and ploidy were estimated with FACETS (version 0.6.2)[99] with the recommended parameter "cval = 400".

Single-base substitutions, excluding repeat regions, were categorized into trinucleotide contexts with SigprofilerMatrixGenerator (version 1.2.12)[100], and de novo signatures were extracted with SigprofilerExtracter (version 1.1.2)[101,102] using nonnegative matrix factorization (NMF) with default parameters. These signatures were decomposed to COSMIC signatures (version 3.3) using cosine similarity and assigned to each sample.

## Statistical analysis

Boxplot data are presented as the biological means ± SEM for biological repetition or ± SD for technical repetition. For two-group experiments, statistical significance was determined by a $t$-test (see the figure legends). For multiple-group experiments, statistical significance was determined by multiple comparison test (see the figure legends). All statistical analysis were performed using Prism 9 software, with $p < 0.05$ considered statistically significant.

## Reporting summary

Further information on research design is available in the Nature Portfolio Reporting Summary linked to this article.

## Data availability

RNA-seq data from SAEC cell samples generated in this study have been deposited in the NCBI GEO database under accession code "GSE223027". ChIP-seq data from SAEC cell samples used in this study have been deposited in the DDBJ data base under accession code "DRA002311". Proteomics data generated in this study have been deposited in JPOST, a public proteome database certified by the ProteomeXchange Consortium, under accession code "PXD043419 [https://repository.jpostdb.org/preview/1810070348649e9ff3ac6da] (Access key:2101)". WGS data generated in this study have been deposited in DDBJ under accession code "PRJDB16238". UniProt human database is available at (https://www.uniprot.org). Human genome (GRCh38) data is available at (https://gdc.cancer.gov/about-data/gdc-data-processing/gdc-reference-files).The Whole Transcriptome Sequencing (WTS) data from NCCJ-cohort that support the findings of this study are not publicly available and restrictions apply to the availability of these data. Such WTS data are available through to the corresponding authors (Bunsyo Shiotani: bshiotan@ncc.go.jp) for academic non-commercial research purposes upon reasonable request, and subject to review of a project proposal that will be evaluated by PRISM data access committee, entering into an appropriate data access agreement and subject to any applicable ethical approvals. The data supporting the findings of this study are available within the Article and its supplementary information files or Source Data file. Source data are provided with this paper.

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

## Acknowledgements

We thank Dr. Tohru Kiyono for the retroviral plasmid constructs and the SAECs, Dr. Helfrid Hochegger for the RPE-1-puro cells, Dr. Yoshihisa Kobayashi for the H3122 cells, Dr. Makoto Nakanishi for the lentivirus plasmid constructs, Dr. Yusuke Yamamoto for the helpful advice on RNA-seq analysis, and members of the Shiotani lab for discussions. This work was supported by the Princess Takamatsu Cancer Research Fund (17-24912), the National Cancer Center Research and Development Fund (29-Seeds-4), and JSPS

KAKENHI (18KK0235 and 18H03378) to B.S. and in part by the Japan Science and Technology Agency (JST) CREST (JPMJCR1689) to R.H. and AIP-PRISM (JPMJCR18Y4) to R.H.

## Author contributions

T.I., M.M., and B.S. designed the project. T.I., M.M., and K.Y. performed the cellular and biochemical experiments and data analysis. T.I., T.Ya., M.N., T. Yo., Y.Y., K.S., H.H., T.K., R.H., and B.S. performed clinical data analysis. T.I. and J.A. performed proteomic experiments and data analysis. T.I., A.M., K.S., T.K., R.H., and B.S. performed whole-genome sequencing experiments and data analysis. N.T., A.Y., L.Z., and B.S. supervised the experiments and data analysis. T.I. and B.S. wrote the manuscript with contributions from all other authors.

## Competing interests

The authors declare no competing interests.

## Additional information

[1]Laboratory of Genome Stress Signaling, National Cancer Center Research Institute, Chuo-ku, Tokyo 104-0045, Japan. [2]Department of Biosciences, School of Science, Kitasato University, Minami-ku, Sagamihara-city, Kanagawa 252-0373, Japan. [3]Department of Late Effects Studies, Radiation Biology Center, Graduate School of Biostudies, Kyoto University, Sakyo-ku, Kyoto 606-8501, Japan. [4]Division of Genome Biology, National Cancer Center Research Institute, Chuo-ku, Tokyo 104-0045, Japan. [5]Department of Respiratory Medicine, Tokyo Medical and Dental University, Bunkyo-ku, Tokyo 113-8519, Japan. [6]Department of Thoracic Oncology, National Cancer Center Hospital, Chuo-ku, Tokyo 104-0045, Japan. [7]Department of Thoracic Surgery, National Cancer Center Hospital, Chuo-ku, Tokyo 104-0045, Japan. [8]Division of Cancer RNA Research, National Cancer Center Research Institute, Chuo-ku, Tokyo 104-0045, Japan. [9]Department of Clinical Genomics, National Cancer Center Research Institute, Chuo-ku, Tokyo 104-0045, Japan. [10]Division of Medical AI Research and Development, National Cancer Center Research Institute, Chuo-ku, Tokyo 104-0045, Japan. [11]Laboratory of Proteomics for Drug Discovery, Laboratory of Clinical and Analytical Chemistry, Center for Drug Design Research, National Institutes of Biomedical Innovation, Health and Nutrition, Ibaraki-city, Osaka 567-0085, Japan. [12]Massachusetts General Hospital Cancer Center, Harvard Medical School, Charlestown, MA 02129, USA. [13]Department of Pathology, Massachusetts General Hospital, Harvard Medical School, Boston, MA 02115, USA. [14]Department of Pharmacology and Cancer Biology, Duke University School of Medicine, Durham, NC 27708, USA. ✉e-mail: bshiotan@ncc.go.jp

