## [Peer Review File · Nature Communications]

An ATR-PrimPol pathway confers tolerance to oncogenic KRAS-induced and heterochromatin-associated replication stressEditorial Note: This manuscript has been previously reviewed at another journal that is not operating a transparent peer review scheme. This document only contains reviewer comments and rebuttal letters for versions considered at *Nature Communications*. Mentions of the other journal have been redacted.

REVIEWERS' COMMENTS

Reviewer #1 (Remarks to the Author):

The authors have revised the manuscript according to the comments made for the original version of the manuscript. This is a well-done study reporting important insights on how cells cope with oncogene-induced DNA replication stress. The authors provide new data on the effects of chloroquine and on chromatin compaction. The current version of the manuscript is suitable for publication.

Reviewer #2 (Remarks to the Author):

Igarashi et al. presented a substantially improved version of their manuscript originally submitted to [REDACTED]. I have still some concerns, nevertheless, the authors answered satisfactorily most of my comments. The new results presented in Fig. 7 and S7 answer one of my main previous concerns, which definitely made the manuscript more compelling.

I am not fully convinced by some of the arguments used by the authors addressing some of my concerns, that does not mean that the story is not publishable, on the contrary, the manuscript could be accepted with some minor changes. I leave the readers of NCOMM to enjoy and judge this interesting paper.

Minor remarks,

Supplementary Figure S1 panel E, statistical analysis and n number could be added.

Supplementary Figure S5 panel B, the authors should add two missing conditions in their analysis, siHP1#1 without 4OHT and siHP1#3 without 4OHT. These controls are important to show that there is, or is not, any effect of the knockdowns on RPA intensity. It could be also interesting for the readers to see the analysis of RPA-foci formation in the same experimental settings.

Not addressed in this version and upon editorial consideration, the discussion is unnecessarily long, and some parts are redundant in describing again their results.

We thank the reviewers and the editor for their comments on our manuscript. We are delighted to find that Reviewer 1 has now supported the publication of the revised manuscript. In this round of review, Reviewer 2 brought up additional comments, which we addressed in the revision with additional experiments, and shortening discussion. We believe we have addressed all remaining remarks to the best of our ability.

Reviewer #1 (Remarks to the Author):

The authors have revised the manuscript according to the comments made for the original version of the manuscript. This is a well-done study reporting important insights on how cells cope with oncogene-induced DNA replication stress. The authors provide new data on the effects of chloroquine and on chromatin compaction. The current version of the manuscript is suitable for publication.

We thank Reviewer#1 for suggestions and support that helped us improve our manuscript.

Reviewer #2 (Remarks to the Author):

Igarashi et al. presented a substantially improved version of their manuscript originally submitted to [REDACTED]. I have still some concerns, nevertheless, the authors answered satisfactorily most of my comments. The new results presented in Fig. 7 and S7 answer one of my main previous concerns, which definitely made the manuscript more compelling.

I am not fully convinced by some of the arguments used by the authors addressing some of my concerns, that does not mean that the story is not publishable, on the contrary, the manuscript could be accepted with some minor changes. I leave the readers of NCOMM to enjoy and judge this interesting paper.

Thank you for your understanding and comments. As shown below, we conducted experiments to support our conclusions. We believe that all of your concerns are addressed by the revision.

Minor remarks,

1. Supplementary Figure S1 panel E, statistical analysis and n number could be added.

We have repeated experiments in Figure 1e and shown in the revised Supplementary Figure 1e (top panel) although the sample of control cells treated with 0.1 μ M of 4OHT for 21 days is missing. In these two experiments, ATR expression reproducibly started to increase on day 14 and remained higher than the basal level until day 28. Statistical analysis in Supplementary Figure 1e (bottom panel) showed that ATR expression was significantly higher in cells with 4OHT than in cells without.

2. Supplementary Figure S5 panel B, the authors should add two missing conditions in their analysis, siHP1#1 without 4OHT and siHP1#3 without 4OHT. These controls are important to show that there is, or is not, any effect of the knockdowns on RPA intensity. It could be also interesting for the readers to see the analysis of RPA-foci formation in the same experimental settings.

We thank the reviewer for raising this important point. We repeated experiments in Supplementary Figure 5b with the controls, siHP1#1 without 4OHT and siHP1#3 without 4OHT. In our new experiment, KRAS^{G12V} expression reproducibly increased chromatin-bound RPA, and this increase was suppressed by HP1 α depletion with siHP1#1 and #3. Notably, siHP1#1 suppressed chromatin-bound RPA slightly more efficiently than siHP1#3, possibly because siHP1#1 depletes HP1 α more efficiently. In the absence of 4OHT, siHP1#1 also slightly reduced chromatin-bound RPA, suggesting that some of the baseline replication stress in cells without KRAS^{G12V} is also heterochromatin-dependent. Nonetheless, siHP1#3 did not reduce chromatin-bound RPA significantly, which is likely a result of low baseline replication stress and inadequate HP1 α knockdown. Overall, our new results confirmed our previous data and directly addressed the reviewer's concern on whether knockdown affects RPA intensity.

3. Not addressed in this version and upon editorial consideration, the discussion is unnecessarily long, and some parts are redundant in describing again their results.

In accordance with the reviewer's remarks, redundant statements regarding the results were removed and the "Discussion" was shortened.